# Align-DA: Align Score-based Atmospheric Data Assimilation with Multiple Preferences

**Jing-An Sun**[*]
Fudan University
Shanghai Artificial Intelligence Laboratory
jasun22@m.fudan.edu.cn

**Hang Fan**[*]
Columbia University
hf2526@columbia.edu

**Junchao Gong**
Shanghai Artificial Intelligence Laboratory
Shanghai Jiaotong University
gjchimself@sjtu.edu.cn

**Ben Fei**[†]
The Chinese University of Hong Kong
Shanghai Artificial Intelligence Laboratory
benfei@cuhk.edu.hk

**Kun Chen**
Fudan University
Shanghai Artificial Intelligence Laboratory
kunchen22@m.fudan.edu.cn

**Fenghua Ling**
Shanghai Artificial Intelligence Laboratory
lingfenghua@pjlab.org.cn

**Wenlong Zhang**
Shanghai Artificial Intelligence Laboratory
zhangwenlong@pjlab.org.cn

**Wanghan Xu**
Shanghai Artificial Intelligence Laboratory
Shanghai Jiaotong University
xu_wanghan@sjtu.edu.cn

**Li Yan**
Fudan University
cli.yan@outlook.com

**Pierre Gentine**
Columbia University
pg2328@columbia.edu

**Lei Bai**
Shanghai Artificial Intelligence Laboratory
baisanshi@gmail.com

## Abstract

Data assimilation (DA) aims to estimate the full state of a dynamical system by combining partial and noisy observations with a prior model forecast, commonly referred to as the background. In atmospheric applications, this problem is fundamentally ill-posed due to the sparsity of observations relative to the high-dimensional state space. Traditional methods address this challenge by simplifying background priors to regularize the solution, which are empirical and require continual tuning for application. Inspired by alignment techniques in text-to-image diffusion models, we propose Align-DA, which formulates DA as a generative process and uses reward signals to guide background priors—replacing manual tuning with data-driven alignment. Specifically, we train a score-based model in the latent space to approximate the background-conditioned prior, and align it

---

[*]Equal contribution.
[†]Corresponding author.

using three complementary reward signals for DA: (1) assimilation accuracy, (2) forecast skill initialized from the assimilated state, and (3) physical adherence of the analysis fields. Experiments with multiple reward signals demonstrate consistent improvements in analysis quality across different evaluation metrics and observation-guidance strategies. These results show that preference alignment, implemented as a soft constraint, can automatically adapt complex background priors tailored to DA, offering a promising new direction for advancing the field.

## 1 Introduction

Data assimilation (DA) estimates the posterior state of a dynamical system by integrating prior model forecasts $\boldsymbol{x}_b$ (background) with observations $\boldsymbol{y}$ [1, 2, 3]. However, in practical applications, the number of observations is often much smaller than the dimensionality of the model state, making DA an underdetermined probabilistic problem—multiple plausible states can equally explain the same set of observations. This inherent ambiguity raises a fundamental question: what defines a good analysis for DA, and how can it be reliably obtained in the face of such uncertainty?

In atmospheric science, data assimilation (DA) provides accurate initial conditions that are essential for reliable weather forecasting. It is also used to generate high-fidelity reanalysis datasets, which serve as a fundamental resource for climate research [1, 2, 4]. The quality of atmospheric analysis is commonly evaluated based on three criteria: accuracy, downstream forecast performance, and adherence to physical constraints. Over decades of development, modern atmospheric DA has adopted a Bayesian inference framework, typically modeling the background-conditioned prior $p(\boldsymbol{x}|\boldsymbol{x}_b)$ as a Gaussian distribution with empirically constructed covariance [5, 3, 6, 7, 8]. To ensure the analysis satisfies these criteria, operational systems depend on extensive parameter tuning-adjusting correlation length scales in variational methods [9], calibrating localization and inflation in ensemble Kalman filters [10], and modifying ensemble weights in hybrid approaches [11, 12]. While these traditional approaches have proven effective, they are often computationally inefficient and labor-intensive. The remarkable success of machine learning in medium-range weather prediction [13, 14, 15, 16, 17] has spurred a growing interest in ML-based DA [18, 19, 20, 21, 22] as a core component of end-to-end forecasting systems [23, 24, 25]. However, current ML approaches face a fundamental limitation: they lack explicit mechanisms to incorporate domain-specific knowledge, especially physical constraints [25, 26, 27]. While some studies have applied reinforcement learning to data assimilation for the above challenge, these approaches typically frame DA as a sequential decision-making problem [28, 29, 30], which is challenging for direct application to large-scale, high-dimensional systems like global weather [31, 32].

To this end, we propose **Align-DA**, a novel framework that aligns DA analyses with key preferences in atmospheric applications using reinforcement learning. It is the introduction of a new paradigm shift for DA, replacing the manual, experience-driven tuning process with a preference alignment-based framework, which becomes the core contribution of this paper. Instead of a new predictive model architecture, we propose a pluggable alignment module designed to enhance any score-based DA method, enabling it to satisfy complex domain preferences that are difficult to model directly. Specifically, we model the background-conditioned prior distribution $p(\boldsymbol{x}|\boldsymbol{x}_b)$ directly using a diffusion model and incorporate observations via guided sampling to approximate the posterior $p(\boldsymbol{x}|\boldsymbol{x}_b,\boldsymbol{y})$ [33]. Inspired by alignment techniques in diffusion-based image generation, we further refine the pretrained diffusion model via direct preference optimization (DPO) [34] using domain-specific reward signals tailored for atmospheric DA. As illustrated in Figure 1, this alignment process makes the posterior $p(\boldsymbol{x}|\boldsymbol{x}_b,\boldsymbol{y})$ more focused, effectively narrowing the solution space and enabling better analysis and forecast performance.

Given the high dimensionality of global atmospheric states, we implement the Align-DA framework in a latent space learned via a variational autoencoder (VAE). Here, we consider three complementary reward metrics: assimilation accuracy, forecast skill, and physical adherence. Our experiments reveal two critical insights: 1) Joint optimization across all reward signals leads to consistent improvement in various score-based DA implementations, and 2) adjusting the reward composition allows the DA analysis to adapt to different downstream tasks. These results demonstrate the effective integration of prior knowledge through **soft-constrain** DPO, which opens a new avenue for enhancing DA. Moreover, our framework supports flexible reward design, making it adaptable to a wide range of DA scenarios. Our contributions are outlined as follows,

- **Novel RL-driven DA framework**: We propose Align-DA, a pioneering reinforcement learning-based DA framework that replaces empirical tuning with preference alignment and consistently improves analysis and forecast performance across score-based methods.
- **Soft constrain DA formalism**: The proposed framework enables the incorporation of implicit or hard-to-model constraints (e.g., forecast performance, geostrophic balance) through soft, reward-based alignment, offering a flexible new avenue for enhancing DA.

## 2 Related work

**Traditional data assimilation.** Traditional DA methods are fundamentally grounded in Bayesian inference, aiming to estimate the posterior distribution $p(\boldsymbol{x}|\boldsymbol{x}_b, \boldsymbol{y})$, which is proportional to $p(\boldsymbol{y}|\boldsymbol{x})p(\boldsymbol{x}|\boldsymbol{x}_b)$. The most widely used variational and ensemble Kalman filter methods typically assume that both the prior $p(\boldsymbol{x}|\boldsymbol{x}_b)$ and observation errors $p(\boldsymbol{y}|\boldsymbol{x})$ follow Gaussian distributions [3, 5, 6]. Taking variational methods as an example, the analysis state is obtained by maximizing the posterior probability $p(\boldsymbol{x}|\boldsymbol{x}_b)p(\boldsymbol{y}|\boldsymbol{x})$, which is equivalent to minimizing the following cost function in a three-dimensional DA scenario:

$$J(\boldsymbol{x}) = \frac{1}{2}(\boldsymbol{x} - \boldsymbol{x}_b)^T \mathbf{B}^{-1}(\boldsymbol{x} - \boldsymbol{x}_b) + \frac{1}{2}(\boldsymbol{y} - \mathcal{H}\boldsymbol{x})^T \mathbf{R}^{-1}(\boldsymbol{y} - \mathcal{H}\boldsymbol{x}) \,. \tag{1}$$

where $\mathbf{B}$ and $\mathbf{R}$ denote the background and observation error covariance matrices, respectively. $\mathcal{H}$ represents the observation operator. For numerical weather prediction systems, the dimensionality of $\mathbf{B}$ can reach $10^{14}$, making its estimation and storage computationally intractable. Consequently, $\mathbf{B}$ is often simplified and empirically tuned based on the quality of the resulting analysis and the forecasts initialized from it. Recent advances in latent data assimilation (LDA) [35, 36, 37, 38] enable more efficient assimilation in a compact latent space. LDA also alleviates the challenge of estimating $\mathbf{B}$, but still relies on empirical tuning.

**Score-based data assimilation (SDA).** The emergence of diffusion models [39, 40, 41] has enabled new approaches to data assimilation through their ability to approximate complex conditional distributions $p(\boldsymbol{x}|\boldsymbol{x}_b, \boldsymbol{y})$. Existing approaches primarily differ in how they incorporate the background state and observations into the diffusion process. Rozet et al. [42] and Manshausen et al. [43] treat observations as guidance signals during the reverse diffusion process but omit the background prior, which may lead to suboptimal performance in scenarios with sparse or noisy observations. Huang et al. [33] train a diffusion model conditioned on the background state and incorporate observations via a repainting scheme during sampling; however, this approach struggles to handle complex, nonlinear observation operators such as satellite radiative transfer. Qu et al. [44] encode both the background and observation into a joint guidance signal, but their method is restricted to specific observation distributions, limiting its applicability in general DA settings. Furthermore, current approaches leave two critical challenges unaddressed: (i) they lack explicit mechanisms to enforce assimilation accuracy and forecast skill during optimization, and (ii) the incorporation of physical law remains unexplored.

**The alignment of the score-based model.** Our work presents the first attempt to enhance prior knowledge encoding in score-based data assimilation and embed the physical constraints through preference alignment in a soft style, drawing inspiration from recent advances in diffusion model optimization. In the text-to-image domain, following the success of RLHF in language models [45], recent work [46, 47, 48] has demonstrated the potential of fine-tuning diffusion models via reward models that capture human preferences [49, 50, 51, 52, 53]. Seminal works by Black et al. [46] and Fan et al. [49] pioneered the formulation of discrete diffusion sampling as reinforcement learning problems, enabling policy gradient optimization, though their approaches faced challenges with computational efficiency and training stability. The field advanced significantly with Diffusion-DPO [54], which adapted Direct Preference Optimization [55] to align diffusion models with human preferences through image pairs. Subsequent innovations like DSPO [56] introduced direct score function correction, while remaining limited to single-preference alignment. Most recently, breakthroughs such as Capo [57] and BalanceDPO [58] have established multi-preference alignment frameworks - though their application to data assimilation remains unexplored. In the DA context, evaluations traditionally focus on three key metrics: assimilation accuracy, forecast skill, and physical adherence of analyses. Our **Align-DA** framework uniquely bridges these domains by extending and adapting multi-preference alignment techniques to simultaneously optimize this triad of DA objectives through differentiable reward signals.

# 3 Method

## 3.1 The score-based model and data assimilation

The score-based model employs forward and reverse stochastic differential equations (SDEs) [40]. The forward SDE progressively adds noise to data distribution $p_0(\boldsymbol{x})$ through:

$$d\boldsymbol{x} = \mathbf{f}(\boldsymbol{x}, t)dt + g(t)d\boldsymbol{w}, \tag{2}$$

while the reverse-time SDE removes noise:

$$d\boldsymbol{x} = [\mathbf{f}(\boldsymbol{x}, t) - g(t)^2 \nabla_{\boldsymbol{x}} \log p_t(\boldsymbol{x})]dt + g(t)d\bar{\boldsymbol{w}}. \tag{3}$$

Here, $\mathbf{f}(\boldsymbol{x}, t)$ and $g(t)$ control the drift and diffusion, with $\boldsymbol{w}, \bar{\boldsymbol{w}}$ both being the standard Wiener processes. The perturbation kernel follows $p(\boldsymbol{x}_t|\boldsymbol{x}) \sim \mathcal{N}(\mu(t), \sigma^2(t)\boldsymbol{I})$, where we adopt a variance-preserving SDE with cosine scheduling schedule for $\mu(t)$ [42]. The score function $\nabla_{\boldsymbol{x}} \log p_t(\boldsymbol{x})$ can be estimated by a neural network $\boldsymbol{s_\theta}(\boldsymbol{x}, t)$ via minimizing the denoising score matching loss $\mathcal{L}_t \equiv \mathbb{E}_{p(\boldsymbol{x}_t)} ||\boldsymbol{s_\theta}(\boldsymbol{x}, t) - \nabla_{\boldsymbol{x}} \log p_t(\boldsymbol{x}|\boldsymbol{x}_0)||^2$ [40]. Once trained, samples are generated by solving the reverse SDE using the learned score estimate $\boldsymbol{s_\theta}(\boldsymbol{x}, t) \approx \nabla_{\boldsymbol{x}} \log p_t(\boldsymbol{x})$.

In SDA, we aim to sample the analysis field through $\boldsymbol{x} = \arg\max_{\boldsymbol{x}} p(\boldsymbol{x}|\boldsymbol{x}_b, \boldsymbol{y})$. Following the variational DA approaches, the posterior score function can be decomposed into two components:

$$\nabla_{\boldsymbol{x}_t} \log p(\boldsymbol{x}_t|\boldsymbol{x}_b, \boldsymbol{y}) = \nabla_{\boldsymbol{x}_t} \log p(\boldsymbol{x}|\boldsymbol{x}_b) + \nabla_{\boldsymbol{x}_t} \log p(\boldsymbol{y}|\boldsymbol{x}_t)$$
$$= \boldsymbol{s_\theta}(\boldsymbol{x}_t, \boldsymbol{x}_b) + \nabla_{\boldsymbol{x}_t} \log p(\boldsymbol{y}|\boldsymbol{x}_t). \tag{4}$$

The first term represents a conditional score function of $p(\boldsymbol{x}|\boldsymbol{x}_b)$, leveraging the expressiveness of diffusion models to learn complex prior distributions of analysis fields directly from data. The observation-associated score function is called guidance, enforcing observational constraints. In DiffDA [33], the repainting technique is used to approximate this guidance. While DPS Guidance [34] retains the observation Gaussian likelihood assumption in variational DA: $p(\boldsymbol{y}|\boldsymbol{x}_t) \sim \mathcal{N}(\boldsymbol{y}|\mathcal{H}(\hat{\boldsymbol{x}}_0(\boldsymbol{x}_t)), \boldsymbol{R})$, where

$$\hat{\boldsymbol{x}}_0(\boldsymbol{x}_t) = \frac{\boldsymbol{x}_t + \sigma(t)^2 \nabla_{\boldsymbol{x}_t} \log p(\boldsymbol{x}_t|\boldsymbol{x}_b)}{\mu(t)} \approx \frac{\boldsymbol{x}_t + \sigma(t)^2 \boldsymbol{s_\theta}(\boldsymbol{x}_t, \boldsymbol{x}_b)}{\mu(t)} \tag{5}$$

is the posterior mean given by Tweedie's formula diffusion model [41].

## 3.2 The observation guidance methods in latent space

Direct SDA in high-dimensional atmospheric model space poses significant computational challenges. To overcome this, we employ VAE to compress physical fields into low-dimensional manifolds [59, 60] and perform data assimilation within this reduced latent space. Our framework first trains a score-based model to learn the background conditional prior $p(\boldsymbol{z}|\boldsymbol{z}_b)$. For observation integration, we adapt two guidance approaches to the latent space.

The score function decomposition forms the foundation for our latent DPS guidance approach:

$$\nabla_{\boldsymbol{z}_t} \log p(\boldsymbol{z}_t|\boldsymbol{z}_b, \boldsymbol{y}) \approx \boldsymbol{s_\theta}(\boldsymbol{z}_t, \boldsymbol{z}_b) + \nabla_{\boldsymbol{z}_t} \log p(\boldsymbol{y}|\boldsymbol{z}_t), \tag{6}$$

where the observation term is assumed Gaussian $p(\boldsymbol{y}|\boldsymbol{z}_t) \sim \mathcal{N}(\mathcal{H}(D(\hat{\boldsymbol{z}}_0(\boldsymbol{z}_t)), \boldsymbol{R})$. Similar to Equation 5, $\hat{\boldsymbol{z}}_0 = (\tilde{\boldsymbol{z}}_t + \sigma^2(t) \nabla_{\boldsymbol{z}_t} \log p(\boldsymbol{z}_t|\boldsymbol{z}_b))/\mu(t)$ is the posterior mean. The $D(\cdot)$ denotes the decoder of VAE. Assuming the $\tilde{\boldsymbol{z}}_t$ is the denoised latent at reverse time $t$, latent counterpart of DPS guidance is formulated as:

$$\boldsymbol{z}_t = \tilde{\boldsymbol{z}}_t + \zeta \nabla_{\boldsymbol{z}_t} \log p(\boldsymbol{y}|\boldsymbol{z}_t)$$
$$= \tilde{\boldsymbol{z}}_t - \frac{1}{2}\zeta \nabla_{\tilde{\boldsymbol{z}}_t}(\boldsymbol{y} - \mathcal{H}(D(\hat{\boldsymbol{z}}_0)))^T \boldsymbol{R}^{-1}(\boldsymbol{y} - \mathcal{H}(D(\hat{\boldsymbol{z}}_0))), \tag{7}$$

where $\zeta$ is the guidance scale. We also implement a latent-space adaptation of Repaint techniques [33]. We first sample the observation-informed latent $\boldsymbol{z}_t^{obs} \sim \mathcal{N}(\mu(t)E(\boldsymbol{x}), \sigma^2(t)\boldsymbol{I})$ with noise, where $\boldsymbol{x}$ is the ERA5 truth state. Then, we fuse the observation information and background prior through masked composition in the physical space:

$$\boldsymbol{z}_{t-1} = E\left(m \odot D(\boldsymbol{z}_t^{obs}) + (1 - m) \odot D(\tilde{\boldsymbol{z}}_t)\right), \tag{8}$$

where $m$ is a spatial mask indicating observed regions and $E(\cdot)$ is the VAE encoder.

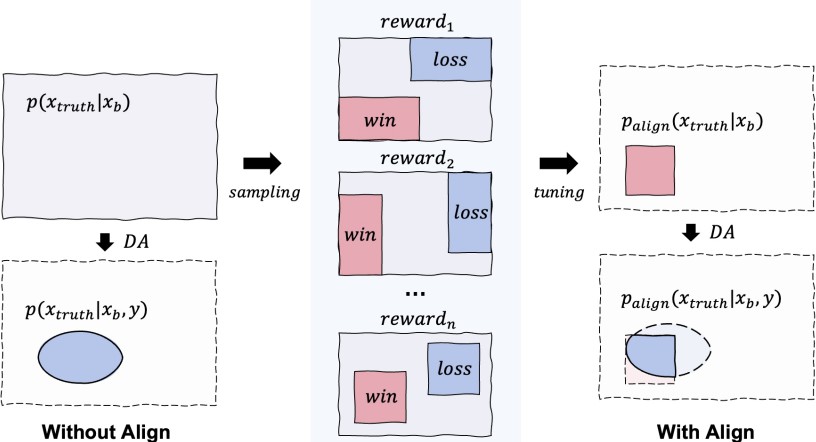

Figure 1: **Schematic of Align-DA.** Conventional score-based DA learns a background-conditioned diffusion model to approximate a broad prior distribution $p(\boldsymbol{x}_{\text{truth}} \mid \boldsymbol{x}_b)$, which, after incorporating observations, yields a posterior estimate $p(\boldsymbol{x}_{\text{truth}} \mid \boldsymbol{x}_b, \boldsymbol{y})$ that may be overly dispersed (left). Align-DA leverages reward-guided alignment to adaptively refine and concentrate the prior $p_{\text{align}}(\boldsymbol{x}_{\text{truth}} \mid \boldsymbol{x}_b)$, resulting in a posterior $p_{\text{align}}(\boldsymbol{x}_{\text{truth}} \mid \boldsymbol{x}_b, \boldsymbol{y})$ that better reflects observational constraints and more closely aligns with the requirements of DA (right).

### 3.3 The diffusion preference optimization.

The preliminary stage of SDA establishes conditional priors for potential analysis fields. However, empirical evidence reveals substantial deviations between these prior estimates and the true analysis fields [33, 43], indicating significant room for prior knowledge refinement. Furthermore, traditional data assimilation approaches and existing AI-assisted variants face inherent challenges in enforcing physical adherence constraints through conventional knowledge embedding [18, 19, 20, 21]. Recognizing these dual challenges of unreliable prior knowledge estimation and inadequate physical adherence enforcement, we propose a novel **soft-constraint framework** on the basis of diffusion preference optimization (Diffusion-DPO) [54], extending its recent success in text-to-image generation to atmospheric data assimilation.

The reward finetuning, seeking to maximize the expected reward $r(\boldsymbol{x}, c)$ while maintaining proximity to the reference distribution $p_{\text{ref}}$ via KL regularization [61]:

$$\max_{\theta} \mathbb{E}_{c, x_0 \sim p_\theta}[r(\boldsymbol{x}_0, c)] - \beta D_{\text{KL}}(p_\theta(\cdot|c) \parallel p_{\text{ref}}(\cdot|c)), \tag{9}$$

where $\beta$ modulates the regularization strength. Following the Bradley-Terry preference model [62, 45], one can construct the reward signal through pairwise comparisons:

$$p_{\text{BT}}(\boldsymbol{x}_0^w \succ \boldsymbol{x}_0^l | c) = \sigma(r(\boldsymbol{x}_0^w, c) - r(\boldsymbol{x}_0^l, c)), \tag{10}$$

with the reward model trained via negative log-likelihood minimization:

$$\mathcal{L}_{\text{BT}}(r) = -\mathbb{E}(c, \boldsymbol{x}_0^w, \boldsymbol{x}_0^l) \left[ \log \sigma(r(c, \boldsymbol{x}_0^w) - r(c, \boldsymbol{x}_0^l)) \right], \tag{11}$$

where $\boldsymbol{x}_0^w$ and $\boldsymbol{x}_0^l$ are the generated "winning" and "losing" samples with condition $c$. In this work, the "winning" and "losing" samples are selected across multi-rewards (see details in Section 4.2).

The analytical solution of Equation 9, $p^*(\boldsymbol{x}_0|c) \propto p_{\text{ref}}(\boldsymbol{x}_0|c) \exp\left(\frac{1}{\beta} R(\boldsymbol{x}_0, c)\right)$, yields the direct preference optimization objective:

$$\mathcal{L}(\theta) = -\log \sigma \left( \beta \left[ \log \frac{p_\theta(\boldsymbol{x}_0^w|c)}{p_{\text{ref}}(\boldsymbol{x}_0^w|c)} - \log \frac{p_\theta(\boldsymbol{x}_0^l|c)}{p_{\text{ref}}(\boldsymbol{x}_0^l|c)} \right] \right), \tag{12}$$

making it bypass the reward model training.

To circumvent intractable trajectory-level computations when adopting the DPO for the diffusion model, DiffusionDPO [54] decomposes the trajectory-level DPO loss into the following step-wise

form:

$$\mathcal{L}_{\text{Diff-DPO}} = -\mathbb{E}_{(\boldsymbol{x}_{t-1}^w, \boldsymbol{x}_t^w, \boldsymbol{x}_{t-1}^l, \boldsymbol{x}_t^l)} \left[ \log \sigma \left( \beta \log \frac{p_\theta(\boldsymbol{x}_{t-1}^w \mid \boldsymbol{x}_t^w, c)}{p_{\text{ref}}(\boldsymbol{x}_{t-1}^w \mid \boldsymbol{x}_t^w, c)} - \beta \log \frac{p_\theta(\boldsymbol{x}_{t-1}^l \mid \boldsymbol{x}_t^l, c)}{\pi_{\text{ref}}(\boldsymbol{x}_{t-1}^l \mid \boldsymbol{x}_t^l, c)} \right) \right],$$
(13)

where $\boldsymbol{x}_t^w$ and $\boldsymbol{x}_t^l$ are the noised $\boldsymbol{x}_0^w$ and $\boldsymbol{x}_0^l$, respectively. Finally, through noise prediction network reparameterization, the implementable training objective can be derived:

$$\mathcal{L}(\theta) = -\mathbb{E}_{(x^w, x^l) \sim \mathcal{D} \ t \sim (0,1) \ \epsilon^w, \epsilon^l \sim \mathcal{N}(0, \boldsymbol{I})} \left[ \log \sigma \left( -\beta T \omega(\lambda_t) \left( \Delta_\theta^\epsilon(x_t^w, \epsilon^w) - \Delta_\theta^\epsilon(x_t^l, \epsilon^l) \right) \right) \right], \quad (14)$$

where $\Delta_\theta^\epsilon(x_t, \epsilon) = \|\epsilon - \epsilon_\theta(x_t, t)\|_2^2 - \|\epsilon - \epsilon_{\text{ref}}(x_t, t)\|_2^2$ quantifies the deviation of learned noise patterns from reference behaviors. With such refinement, the diffusion model is aligned to specific preferences.

## 4 Experiments

### 4.1 Experimental Settings and Evaluations

**The background-conditioned model training and sampling** We introduce an integrated approach for background ObsFree distributions modeling that combines latent space learning with ObsFree diffusion processes. The framework employs a two-stage architecture beginning with a transformer-based VAE [63] that reduces input dimensionality by $16\times$ (processing high-resolution field data $69 \times 128 \times 256$ into compressed latent codes $69 \times 32 \times 64$) (see Appendix B for details). The second stage implements an ObsFree latent diffusion model that learns the mapping $p(\boldsymbol{z}|\boldsymbol{z}_b)$ through stacked transformer blocks [64]. The model utilizes (2,2) patch tokenization and integrates ObsFree information from background latent $\boldsymbol{z}_b$ via cross-attention. We adopt a variance-preserving diffusion process [40] with cosine noise scheduling and train the model using a fixed learning rate of $1e - 4$ (batch size 32), observing stable convergence over 100K training iterations. For ObsFree sampling, we implement a hybrid strategy combining 128-step ancestral prediction with two rounds of Langevin-based refinement, enhancing sample fidelity while maintaining computational efficiency [39, 43, 42].

**Experimental setting** Our data assimilation framework employs the FengWu AI forecasting model [65] (6-hour temporal resolution) to generate background fields. The 48-hour background fields are created through 8 auto-regressive 6-hour forecasts initialized from ERA5 data [4] two days prior. Observational data is simulated by applying random masking to ERA5 ground truth with 99% sparsity, approximating realistic satellite observation coverage patterns. The spatial resolution is $1.40625°$ ($128 \times 256$ grid). Here, the FengWu AI forecasting model serves as a fixed (frozen) background generator for all DA methods, including our proposed methods and the baselines. This ensures a fair and controlled comparison.

### 4.2 The preference dataset construction

In the standard practice of RL-based post-training, a preference dataset is typically required [57, 54, 45]. In this paper, our preference dataset is constructed by generating candidate analysis fields latent given background condition from the pre-trained reference model $p(\boldsymbol{z} \mid \boldsymbol{z}_b)$. To address the multi-preference nature of data assimilation—where single-step assimilation accuracy, forecast skill, and physical adherence are distinct objectives—we adopt a holistic evaluation framework. For a batch of $N$-generated analysis latents, each sample is decoded into the analysis field $\boldsymbol{x}_a$ and evaluated across three rewards:

- **Assimilation Accuracy:** The latitude-weighted root mean square error (WRMSE) [66, 67] between $\boldsymbol{x}_a$ and ERA5 ground truth is computed for all 69 variables. The reward for the $n-$th sample is derived as the average pairwise comparison across variables:

$$R_{\text{assim}}(n) = \frac{1}{N - 1} \sum_{j \neq n} \frac{1}{69} \sum_{m=1}^{m=69} p(\boldsymbol{x}_n^m \succ \boldsymbol{x}_j^m), \quad (15)$$

where the Bradley-Terry preference model is applied at variables level $p(\boldsymbol{x}_n^m \succ \boldsymbol{x}_j^m) = \sigma(-\text{wrmse}[n][m] + \text{wrmse}[j][m])$, and $\text{wrmse}[n][m]$ is the WRMSE of $m-$th variable in $n-$th sample.

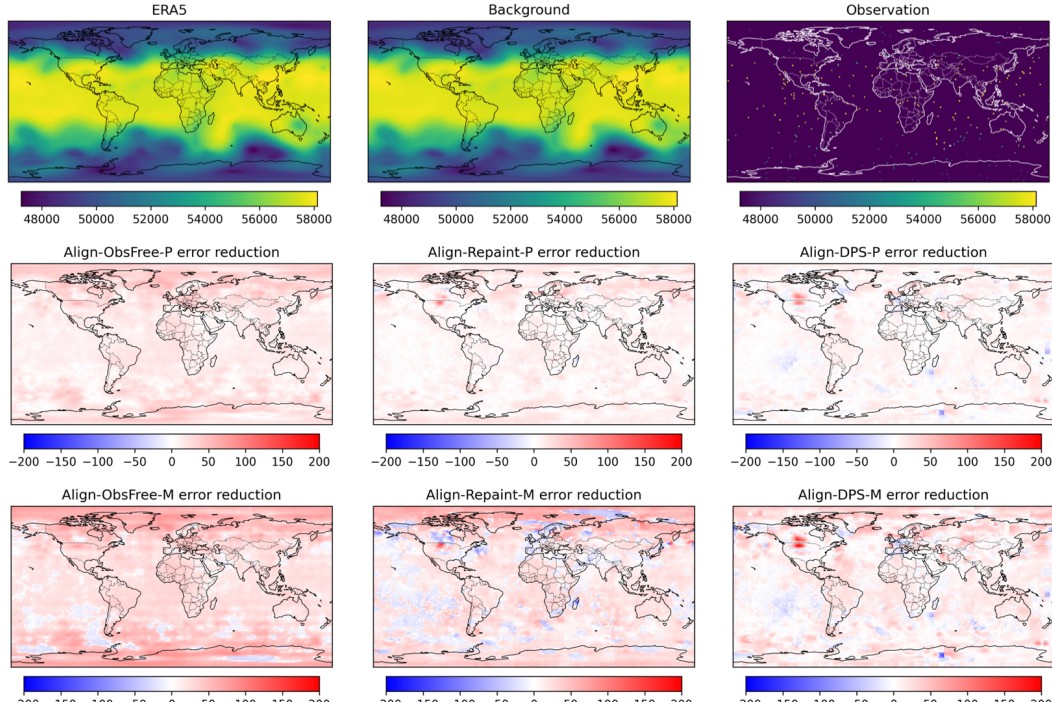

Figure 2: **Visualization of Align-DA Assimilation Performance for Z500 Analysis.** (10 averaged analysis with randomly selected timestamp in 2019) 'ObsFree' means without observation integration baseline. 'Repaint' and 'DPS' represent repaint and diffusion posterior sampling guidance methods. Note that '-P' indicates single physical reward alignment, while '-M' signifies multi-reward alignment. First row: Reference fields showing ERA5 reanalysis (ground truth denoted as GT), background field, and observational data. Second row: Error reduction through single physical reward alignment, quantified as $|\boldsymbol{x}_a^{\mathrm{ref}} - \boldsymbol{x}_{\mathrm{GT}}| - |\boldsymbol{x}_a^{\mathrm{Align-P}} - \boldsymbol{x}_{\mathrm{GT}}|$. Warm hues indicate regions where physical adherence alignment moderately improves accuracy, suggesting potential reward correlations. Third row: Error reduction through multi-reward alignment, quantified as $|\boldsymbol{x}_a^{\mathrm{ref}} - \boldsymbol{x}_{\mathrm{GT}}| - |\boldsymbol{x}_a^{\mathrm{Align-M}} - \boldsymbol{x}_{\mathrm{GT}}|$. More intense warm hues indicate regions where multi-reward alignment substantially improves accuracy compared to the unaligned baseline. The attenuation of color intensity from left to right columns demonstrates diminishing DPO effects as more effective observation guidance methods are applied - a pattern consistent with theoretical expectations.

- **48-hour Forecast Skill:** In this metric, FengWu acts as both an evaluator and a reward source. After each DA method produces an analysis field, we use FengWu to run a 48-hour forecast. The forecast RMSEs relative to ERA5 are computed, yielding a forecast preference score $R_{\mathrm{forecast}}$. A critical distinction of Align-DA is that while baseline methods can be evaluated by FengWu, they lack any mechanism to internalize this feedback to improve their generative process. In contrast, our Align-DA framework is explicitly designed to learn from this feedback, which is the core of its innovation.

- **Physical adherence:** To demonstrate the effectiveness and flexibility of our framework in aligning with physical knowledge, we incorporate reward mechanisms based on two primary examples of atmospheric principles: geostrophic wind balance and hydrostatic balance (see the Appendix F). The geostrophic wind components [68] are defined as:

$$\boldsymbol{u}_g = -\frac{1}{f}\frac{\partial \Phi}{\partial y}, \quad \boldsymbol{v}_g = \frac{1}{f}\frac{\partial \Phi}{\partial x}, \tag{16}$$

where $f$ denotes the Coriolis parameter and $\Phi$ represents the geopotential. We quantify atmospheric field deviations using the metric: $D = \frac{1}{2}\left(\frac{|\boldsymbol{u}-\boldsymbol{u}_g|}{|\boldsymbol{u}|} + \frac{|\boldsymbol{v}-\boldsymbol{v}_g|}{|\boldsymbol{v}|}\right)$. We acknowledge that perfect balance is an idealized approximation. We therefore establish the ERA5 truth deviation $D_{GT}$ as our reference benchmark. The final physical adherence score ($R_{phys}$) for

Table 1: Assimilation accuracy gains of alignment strategies under 1% observation. Percentage values quantify relative accuracy variants compared to the non-aligned baseline for multi-reward experiments.

| | MSE | MAE | WRMSE | | | |
| --- | --- | --- | --- | --- | --- | --- |
| | | | u10 | v500 | z500 | t850 |
| ObsFree | 0.0643 | 0.1382 | 1.5275 | 2.7467 | 104.5531 | 1.0897 |
| Align-ObsFree-P | 0.0645 | 0.1385 | 1.5331 | 2.7501 | 103.1857 | 1.0896 |
| Align-ObsFree-M | 0.0611(−5.44%) | 0.1348(−2.46%) | 1.4942(−2.23%) | 2.7202(−0.97%) | 96.8603(−7.94%) | 1.0866(−0.28%) |
| Repaint | 0.0623 | 0.1368 | 1.5009 | 2.7027 | 103.3036 | 1.0911 |
| Align-Repaint-P | 0.0624 | 0.1369 | 1.5063 | 2.7047 | 101.9634 | 1.0904 |
| Align-Repaint-M | 0.0598(−4.12%) | 0.1344(−1.78%) | 1.4759(−1.69%) | 2.6836(−0.71%) | 96.8906(−6.62%) | 1.0889(−0.20%) |
| DPS | 0.0609 | 0.1337 | 1.4441 | 2.5972 | 93.4654 | 1.1159 |
| Align-DPS-P | 0.0612 | 0.1342 | 1.4514 | 2.6069 | 92.4301 | 1.1184 |
| Align-DPS-M | 0.0593(−2.65%) | 0.1317(−1.51%) | 1.4187(−1.78%) | 2.5800(−0.66%) | 88.4235(−5.70%) | 1.1100(−0.53%) |

Table 2: 48-hour forecast skill gains of alignment strategies under 1% observation.

| | MSE | MAE | WRMSE | | | |
| --- | --- | --- | --- | --- | --- | --- |
| | | | u10 | v500 | z500 | t850 |
| ObsFree | 0.1146 | 0.1918 | 2.1182 | 4.1571 | 232.6243 | 1.4826 |
| Align-ObsFree-P | 0.1151 | 0.1923 | 2.1203 | 4.1623 | 232.5924 | 1.484 |
| Align-ObsFree-M | 0.1080(−6.12%) | 0.1872(−2.47%) | 2.0965(−1.03%) | 4.1091(−1.17%) | 225.7758(−3.03%) | 1.4636(−1.30%) |
| Repaint | 0.1123 | 0.1898 | 2.0899 | 4.1030 | 228.2876 | 1.4672 |
| Align-Repaint-P | 0.1127 | 0.1902 | 2.0912 | 4.1067 | 228.1765 | 1.4677 |
| Align-Repaint-M | 0.1068 (−5.12%) | 0.1860(−2.04%) | 2.0713(−0.90%) | 4.0665(−0.90%) | 222.9336(−2.40%) | 1.4516(−1.07%) |
| DPS | 0.0969 | 0.1757 | 1.9230 | 3.7675 | 193.4167 | 1.3564 |
| Align-DPS-P | 0.0972 | 0.1761 | 1.9275 | 3.7760 | 193.4700 | 1.3599 |
| Align-DPS-M | 0.0943(−2.74%) | 0.1734(−1.30%) | 1.9070(−0.84%) | 3.7287(−1.04%) | 189.6935(−1.96%) | 1.3428(−1.01%) |

a given principle (yielding a *Geo-Score*) is then computed as:

$$R_{phys} = 2 \cdot \sigma \left( -\frac{|D_i - D_{GT}|}{D_{GT}} \right), \tag{17}$$

where $\sigma(\cdot)$ denotes the sigmoid function. This formulation ensures that generated fields closer to the ERA5 ground truth receive higher scores, with the theoretical maximum score of 1 corresponding to perfect agreement with observational data.

To rigorously validate our Align-DA framework, particularly examining the impact of RL-driven physical knowledge embedding and the trial-and-error-free formalism, we construct two distinct preference datasets: a physical adherence reward dataset (i.e. focusing on Geo-Score) and a multi-reward comprehensive set. Both datasets are constructed through systematic sampling of $N = 32$ instances from the pre-trained diffusion model's output space.

For the physical adherence reward dataset, we define winning samples as top-5 performers in Geo-Score, while bottom-5 instances are assigned to the losing group. The comprehensive dataset employs multi-dimensional reward ranking, where winning samples occupy the top decile across all three reward dimensions, with losing counterparts drawn from the bottom decile. To ensure statistical robustness and prevent sampling bias, we implement stratified random pairing with duplicate elimination through hash-based filtering [69]. Our dataset construction leverages 2018 experimental records, yielding a carefully curated 4,000-pair corpus for model fine-tuning. The optimization employs KL regularization ($\beta = 8000$), a batch size of 32, and a learning rate of $1e - 5$, with model checkpoints saved after 500 fine-tuning steps.

## 4.3 Results

We evaluate our framework through two distinct experimental paradigms: single physical adherence reward optimization on the physics reward dataset and multi-reward optimization on the comprehensive dataset. For observation integration, we implement three established methodologies [33, 43]: 1) *ObsFree* - an observation-free baseline model, 2) *Repaint* - the standard repainting technique, also known as inpainting, and 3) *DPS* - diffusion posterior sampling guidance (detailed in Section 3.2). Model variants are denoted through suffix conventions: *-P* indicates single physical reward alignment, while *-M* signifies multi-reward alignment. For instance, *Align-DPS-P*:

Physical adherence reward aligned model with DPS guidance, *Align-ObsFree-M*: Multi-reward aligned model without observation integration, *Repaint*: reference model with repaint integration.

**Posterior analysis improvement.** Table 1 presents quantitative comparisons of posterior analysis accuracy across different alignment approaches. For each observation integration method, we evaluate three configurations: the reference model with no reward alignment, single-physics-reward-aligned model (-P), and multi-reward-aligned model (-M). We report comprehensive metrics including overall MSE, MAE, and variable-specific WRMSE scores, accompanied by percentage improvements relative to the reference baseline.

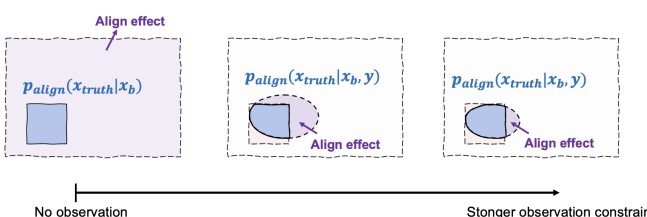

Figure 3: **Interaction between observation constraints and alignment.** Better incorporation of observation information (i.e., stronger constraints) leads to a narrower space for improvement by the alignment process.

Multi-reward alignment consistently outperforms the unaligned model, demonstrating the benefit of directly optimizing for assimilation accuracy. For instance, Align-ObsFree-M reduces overall WRMSE by 5.44%, indicating that multi-objective reward signals lead to tighter posteriors and more accurate reconstructions.

In contrast, single-reward alignment shows mixed results—Align-Repaint-M reduces MSE by 4.12%, but its single-reward version (-P) slightly worsens performance (+0.2%), indicating that physics alone is insufficient for DA tasks. Nevertheless, Align-DPS-P improves Z500 WRMSE across both Repaint and ObsFree settings, illustrating the localized efficacy of enforcing geostrophic balance. Collectively, these findings underscore the importance of reward formulation: while task-specific objectives are essential for global accuracy, domain-specific physical constraints can provide targeted gains. Multi-signal alignment thus offers a more robust and generalizable mechanism for incorporating scientific priors into generative DA.

**Visualization of analysis improvement.** Figure 2 provides a spatial evaluation of reward alignment strategies using Z500 as a diagnostic field. The first row illustrates a representative sample of the ERA5 reanalysis ground truth (GT), background field, and the sparse observations utilized for DA. The second row quantifies error reduction achieved through single physics reward alignment while the third row extends this analysis to multi-reward alignment. The results indicate that the single physical reward consistently improves Z500 across different guidance settings, possibly due to its implicit embedding within the geostrophic wind balance equations. Moreover, the multi-reward strategy yields more substantial improvements than the single reward alone, highlighting the enhanced effectiveness of preference alignment in improving assimilation quality.

**Forecast skill improvement.** Table 2 reports 48-hour forecast performance across different alignment strategies under 1% observation, showing a trend consistent with their effects on analysis. Multi-reward alignment consistently outperforms the baseline across metrics and variables. More importantly, the improvements achieved at analysis time are not only preserved but often amplified in the forecast. For instance, Align-Repaint-M improves analysis MSE by 4.12%, while its 48-hour forecast MSE drops by 5.12%, indicating that aligned states provide more reliable initial conditions for downstream prediction. In contrast, physical-reward-only alignment (–P), which focuses solely on geostrophic balance, continues to yield negligible or even adverse forecast gains. These results highlight the importance of explicitly optimizing for analysis and forecast objectives rather than relying solely on physical priors.

**Alignment benefits fade with strong observational constrain.** The results above demonstrate that, under well-designed reward signals, preference alignment consistently improves both posterior analysis and forecast skill. However, the magnitude of improvement varies across observation incorporation schemes. The observation-free baseline gains the most from alignment, followed by Repaint, while DPS shows the least benefit.

This can be explained by the overlapping effects of observational constraints and preference alignment (Figure 3). The observation-free setting performs sampling without incorporating observational guidance, allowing greater flexibility for reward-based alignment to shape the posterior distribution.

In contrast, DPS imposes the strongest observational constraints—as evidenced by its superior baseline performance in Table 1—leaving limited room for further improvement through alignment.

**Physical adherence improvement.** In this study, we endeavor to enhance physical adherence by employing an alignment technique implemented as a soft constraint, utilizing geostrophic wind balance as a reward signal. Figure 4 presents the Geo-Score, demonstrating that while multi-reward alignment (incorporating physical adherence reward) yields broad improvements across all guidance methods (e.g., from 93.85 to 94.19), optimizing solely with the Geo-Score reward achieves more pronounced enhancements in this specific physical metric (e.g., from 93.85 to 95.00). We acknowledge, however, that a slight degradation in overall assimilation accuracy and forecast skill was observed when only the physical adherence reward was applied. As demonstrated in Appendix E, the performance change of its single-reward version (-P) yields statistically insignificant, sug-

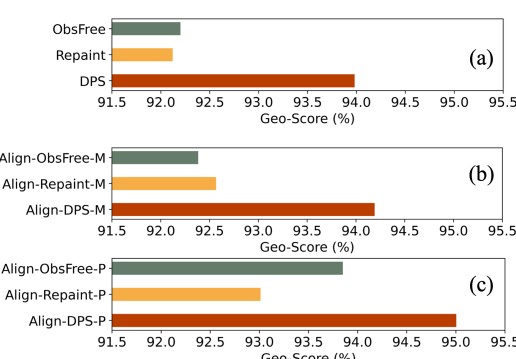

Figure 4: Geo-Score comparisons across alignment strategies: (a) Reference model, (b) Multi-reward aligned (-M), and (c) Single physical adherence reward aligned (-P) results, with colors denoting different guidance methods.

gesting that geostrophic balance alone is insufficient for DA tasks. Nevertheless, the significant Geo-Score improvement substantiates the efficacy of alignment techniques in enhancing physical consistency, highlighting the promising direction of this approach and underscoring the importance of refining physical reward design in future work.

## 5    Conclusion

This work presents **Align-DA**, a novel reinforcement learning framework that introduces **soft-constraint alignment** as a new paradigm for advancing DA. Unlike traditional approaches relying on empirical tuning or hard physical constraints, our method reformulates DA as a preference optimization problem, enabling the flexible integration of domain knowledge through differentiable reward signals. By refining a diffusion-based prior with DPO, the framework effectively narrows the solution space while maintaining adaptability to diverse atmospheric constraints (analysis accuracy, forecast skill, and physical consistency). The error reduction and physical enhancement after alignment demonstrate the efficacy of our alignment framework. While our results are promising, we acknowledge a key limitation of the current approach, which operates in a learned latent space. This is due to the low-dimensional manifold assumption, which makes it inherently difficult for the model to capture rare, outlier events such as extreme weather. Nevertheless, **Align-DA** establishes soft-constraint DPO as a scalable and generalizable alternative to conventional DA formalisms, where implicit or complex prior knowledge (e.g., forecast skill, geostrophic balance) can now be incorporated without restrictive assumptions.

## Acknowledgements

This work is supported by Shanghai Artificial Intelligence Laboratory and the JC STEM Lab of AI for Science and Engineering, funded by The Hong Kong Jockey Club Charities Trust, the Research Grants Council of Hong Kong (Project No. CUHK14213224). This work was done during Jing-An Sun's internship at Shanghai Artificial Intelligence

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

# A Dataset and Evaluations

**Dataset** Our experiments employ the European Centre for Medium-Range Weather Forecasts (ECMWF) atmospheric reanalysis (ERA5) [4] as the primary dataset for model development and validation. The inputs consist of a comprehensive set of meteorological variables, including 5 upper-air atmospheric variables (geopotential $z$, temperature $t$, specific humidity $q$, zonal wind $u$, and meridional wind $v$) with 13 pressure level (50hPa, 100hPa, 150hPa, 200hPa, 250hPa, 300hPa, 400hPa, 500hPa, 600hPa, 700hPa, 850hPa, 925hPa, and 1000hPa), four surface-level measurements (10-meter zonal component of wind (u10), 10-meter meridional component of wind (v10), 2-meter temperature (t2m) and mean sea level pressure (msl)). This configuration yields a total of 69 distinct meteorological variables. We follow ECMWF's standard nomenclature (e.g., q500 for specific humidity at 500 hPa). The temporal coverage spans four decades (1979-2018) to ensure robust model training.

**Evaluations** We implement two standard evaluations for data assimilation quality assessment. (1) Single-step assimilation accuracy: Direct comparison between assimilated states and ERA5 ground truth at current timestep through error metrics. (2) Assimilation-based forecasting: Initializing numerical weather prediction models with assimilated states to compute errors against ERA5 at forecast lead times. Three complementary metrics quantifying performance are overall mean square error (MSE), mean absolute error (MAE), and the latitude-weighted root mean square error (WRMSE), which is a statistical metric widely used in geospatial analysis and atmospheric science. Given the estimate $\hat{x}_{h,w,c}$ and the truth $x_{h,w,c}$, the WRMSE is defined as,

$$\text{WRMSE}(c) = \left[ \frac{1}{HW} \sum_{h,w} H \cdot \left( \frac{\cos \alpha_{h,w}}{\sum_{h'=1}^{H} \cos \alpha_{h',w}} \right) (x_{h,w,c} - \hat{x}_{h,w,c})^2 \right]^{1/2} \tag{18}$$

Here $H$ and $W$ represent the number of grid points in the longitudinal and latitudinal directions, respectively, and $\alpha_{h,w}$ is the latitude of point $(h, w)$.

The validation procedure involves 6-hourly assimilation cycles over the entirety of 2019, where each assimilation employs a 48-hour forecast as its background field. The three designated metrics are calculated for each cycle. Final performance scores are the annual averages of these metrics.

# B The VAE training

We employ the established transformer-based architecture (VAEformer) [63] to compress high-dimensional atmospheric fields into a low-dimensional latent space. This implementation uses window attention [70] to capture atmospheric circulation features effectively. This model follows the "vit_large" configuration for both encoder and decoder, with patch embedding of size (4,4) and stride (4,4), an embedding dimension of 1024, and 24 stacked transformer blocks equipped with window attention. The encoder-decoder architecture is optimized using AdamW [71] with a batch size of 32, trained under a two-phase learning rate policy: a 10,000-step linear warm-up to 2e-4 followed by cosine annealing. The training of VAE is conducted on the ERA5 dataset (1979–2016), with the 2016–2018 period reserved for validation, and runs for a total of 80 epochs. Our trained VAE achieves 0.0067 overall MSE and 0.0486 overall MAE.

# C Computational Costs

The alignment process is computationally efficient. The main costs are divided as follows:

- **One-Time Pre-training:** This is the most intensive step. The VAE training takes approximately 4 days, and the diffusion model pre-training takes about 1 day, both on 4 A100 GPUs.

- **DPO Alignment (Fine-tuning):** This step is highly efficient. The core alignment process completes in about 30 minutes on 4 A100 GPUs after 500 update steps. Reward Evaluation: Evaluating the "forecast skill" for a single data sample involves running a 48-hour forecast, which takes about 4 seconds on one A100 GPU.

- **Inference (Sampling):** Generating a single analysis field takes approximately 30 seconds on one A100 GPU.

This analysis highlights that the novel alignment stage, which provides significant performance gains, adds a minimal computational overhead compared to the initial pre-training.

## D  Comparison with Traditional 3D-Var

To position our work relative to established methods, we conducted a direct comparison against a traditional 3D-Var baseline, a cornerstone of operational weather forecasting. The results are summarized in Table 3.

Table 3: Comparison of Align-DA (Align-DPS-M) with the traditional 3D-Var baseline. While 3D-Var is marginally better on the metric it directly optimizes (DA Accuracy), Align-DA shows significant advantages in the more critical downstream metrics.

| Method | DA Accuracy (MSE) | Forecast Skill (MSE) | Geo-Score |
|---|---|---|---|
| 3D-Var | **0.0581** | 0.0952 | 88.34 |
| Align-DPS-M | 0.0593 | **0.0943** | **94.89** |

The results are highly revealing. While 3D-Var achieves a marginally better DA Accuracy (MSE), which is unsurprising as its cost function is explicitly designed to minimize this value, it falls short in the metrics that are ultimately more critical for weather prediction. Our Align-DA framework demonstrates superior performance in both downstream Forecast Skill and Physical Consistency. The substantial improvement in the Geo-Score (from 88.34 to 94.89) indicates that our method produces analysis fields that are more physically realistic. This superior physical consistency and forecast skill underscore the core value of our contribution: Align-DA's ability to optimize for a complex set of preferences simultaneously, moving beyond the narrow optimization target of traditional methods.

## E  Statistical Significance Testing

To confirm that the performance improvements reported in our paper are statistically significant, we conducted an analysis to quantify the inherent stochasticity of our generative model. We ran the diffusion sampling process 10 times for a single time step under the most uncertain setting—without any observations or alignment ('ObsFree'). The relative standard deviation of the resulting metrics, which represents the model's baseline random variability, is presented in Table 4.

Table 4: Relative standard deviation (%) of key metrics for the unaligned 'ObsFree' baseline, calculated over 10 independent runs. This table quantifies the inherent randomness of the generative process.

| | MSE | MAE | WRMSE | | | |
| | | | u10 | v500 | z500 | t850 |
|---|---|---|---|---|---|---|
| **_DA Accuracy (ObsFree)_** | | | | | | |
| Relative std. dev. (%) | 0.045 | 0.058 | 0.115 | 0.079 | 0.201 | 0.083 |
| **_Forecast Skill (ObsFree)_** | | | | | | |
| Relative std. dev. (%) | 0.163 | 0.067 | 0.145 | 0.155 | 0.252 | 0.106 |

This analysis confirms that our reported improvements are statistically significant. For instance, the relative standard deviation for the primary DA accuracy metric (MSE) is merely **0.045%**. This is orders of magnitude smaller than the **2.65% to 5.44%** performance improvements that our Align-DA framework provides over the baseline (as shown in the main paper). Meanwhile, it is important to note that **the performance change of the single-reward version (-P) yields statistically insignificant results, suggesting that geostrophic balance alone is insufficient for complex DA tasks.**

Table 5: Assimilation accuracy gains with different observation densities. Percentage values quantify relative accuracy variants compared to the non-aligned baseline.

| | | MSE | MAE | WRMSE | | | |
| --- | --- | --- | --- | --- | --- | --- | --- |
| | | | | u10 | v500 | z500 | t850 |
| 1% observation | Repaint | 0.0623 | 0.1368 | 1.5009 | 2.7027 | 103.3036 | 1.0911 |
| | Align-Repaint-M | 0.0598(−4.12%) | 0.3440(−1.78%) | 1.4759(−1.69%) | 2.6836(−0.71%) | 96.8906(−6.62%) | 1.0889(−0.20%) |
| | DPS | 0.0609 | 0.1337 | 1.4441 | 2.5972 | 93.4654 | 1.1159 |
| | Align-DPS-M | 0.0593(−2.65%) | 0.1317(−1.51%) | 1.4187(−1.78%) | 2.5800(−0.66%) | 88.4235(−5.70%) | 1.1100(−0.53%) |
| 3% observation | Repaint | 0.0583 | 0.1327 | 1.4438 | 2.6097 | 100.0290 | 1.0696 |
| | Align-Repaint-M | 0.0560(−4.02%) | 0.1305(−1.73%) | 1.4188(−1.77%) | 2.5929(−0.65%) | 93.8835(−6.54%) | 1.0675(−0.20%) |
| | DPS | 0.0546 | 0.1276 | 1.3698 | 2.4883 | 85.5742 | 1.0586 |
| | Align-DPS-M | 0.0533(−2.53%) | 0.1256(−1.56%) | 1.3457(−1.78%) | 2.4724(−0.65%) | 80.5474(−6.24%) | 1.0558(−0.27%) |
| 5% observation | Repaint | 0.0547 | 0.1290 | 1.3891 | 2.5248 | 96.9590 | 1.0495 |
| | Align-Repaint-M | 0.0526(−3.91%) | 0.1269(−1.68%) | 1.3656(−1.72%) | 2.5081(−0.67%) | 91.0853(−6.45%) | 1.0471(−0.23%) |
| | DPS | 0.0534 | 0.1267 | 1.3597 | 2.4767 | 85.9150 | 1.0514 |
| | Align-DPS-M | 0.0519(−2.78%) | 0.1246(−1.70%) | 1.3337(−1.95%) | 2.4560(−0.84%) | 80.5787(−6.62%) | 1.0475(−0.37%) |
| 10% observation | Repaint | 0.0470 | 0.1204 | 1.2721 | 2.3398 | 90.1747 | 1.0037 |
| | Align-Repaint-M | 0.0453(−3.67%) | 0.1186(−1.52%) | 1.2518(−1.62%) | 2.3230(−0.72%) | 84.7320(−6.42%) | 1.0012(−0.25%) |
| | DPS | 0.0524 | 0.1261 | 1.3520 | 2.4673 | 86.2364 | 1.0468 |
| | Align-DPS-M | 0.0510(−2.83%) | 0.1239(−1.76%) | 1.3252(−2.03%) | 2.4458(−0.88%) | 80.0644(−7.71%) | 1.0403(−0.62%) |

# F   Ablations studies

**Observation densities.** We further investigate the effect of varying observation densities (1%, 3%, 5%, and 10%) on the efficacy of our alignment strategy, as detailed in Table 5. A key finding is that the proposed alignment mechanism significantly enhances assimilation accuracy across all tested densities for both guidance methods. Specifically, Align-Repaint-M and Align-DPS-M consistently outperform their non-aligned counterparts in all scenarios. This benefit is evident even at the highest 10% observation density, where alignment yields substantial error reductions of 6.42% for Align-Repaint-M and 7.71% for Align-DPS-M on the z500 WRMSE metric. While the absolute errors decrease for all methods with more observations, the relative improvement percentage due to alignment shows nuanced behavior. For Align-Repaint-M, the relative gain slightly diminishes as observation density increases (e.g., MSE improvement changes from 4.12% at 1% observations to 3.67% at 10%). This suggests that while alignment is always beneficial, its proportional contribution might be more pronounced when initial information is sparser. Interestingly, Align-DPS-M does not follow a consistent trend. For some metrics like z500 WRMSE, the relative improvement even increases with higher density (e.g., from -5.70% at 1% to -7.71% at 10%). It may be ascribed to the aligned conditional model is easier to be guided in DPS method. Nonetheless, the results affirm the significant and persistent positive impact of the alignment strategy across varying data availability.

**Observation Error Robustness.** Table 6's ablation study examines Align-DPS-M's assimilation accuracy gains over the non-aligned DPS baseline under varying observation error with 1% observation. Results consistently demonstrate the alignment effect's robustness: Align-DPS-M outperforms the DPS baseline across all metrics, even as observation error increases. For example, Align-DPS-M shows a 1.97% MSE reduction and a 4.24% z500 WRMSE reduction even at the highest error (std=0.05). However, while alignment remains beneficial, its relative improvement magnitude over the baseline decreases with increasing observation error. This is evident as Align-DPS-M's percentage gains, though still negative (indicating improvement), diminish in magnitude as observation error rises; for instance, MSE improvement drops from -2.65% ('Idea') to -1.97% (std=0.05), and z500 WRMSE improvement from -5.70% ('Idea') to -4.24% (std=0.05). This suggests that while consistently advantageous, alignment's relative benefit is somewhat attenuated by higher observation noise.

### Alternative Preference Optimization Algorithms

To validate the robustness of our preference alignment framework, we conducted experiments with alternative preference optimization algorithms: Identity Preference Optimisation (IPO) and Direct Preference Optimization with discrete sampling (DSPO). Table 7 presents the comprehensive results for both DA Accuracy and Forecast Skill across our three guidance strategies (ObsFree, Repaint, and DPS). These results confirm that our alignment concept is a powerful and generalizable tool, as both IPO and DSPO also yield significant performance gains over the unaligned baselines.

### Additional Physics Constraints

Table 6: Assimilation accuracy gains with different observation error under 1% observation. Percentage values quantify relative accuracy variants compared to the non-aligned baseline.

| | | MSE | MAE | WRMSE | | | |
| | | | | u10 | v500 | z500 | t850 |
|---|---|---|---|---|---|---|---|
| Idea | DPS | 0.0609 | 0.1337 | 1.4441 | 2.5972 | 93.4654 | 1.1159 |
| | Align-DPS-M | $0.0593_{(-2.65\%)}$ | $0.1317_{(-1.51\%)}$ | $1.4187_{(-1.78\%)}$ | $2.5800_{(-0.66\%)}$ | $88.4235_{(-5.70\%)}$ | $1.1100_{(-0.53\%)}$ |
| std=0.02 | DPS | 0.0607 | 0.1335 | 1.4408 | 2.5959 | 92.5883 | 1.1132 |
| | Align-DPS-M | $0.0593_{(-2.33\%)}$ | $0.1316_{(-1.40\%)}$ | $1.4176_{(-1.64\%)}$ | $2.5805_{(-0.60\%)}$ | $88.5310_{(-4.58\%)}$ | $1.1087_{(-0.41\%)}$ |
| std=0.05 | DPS | 0.0602 | 0.1330 | 1.4358 | 2.5904 | 91.3947 | 1.1056 |
| | Align-DPS-M | $0.0590_{(-1.97\%)}$ | $0.1312_{(-1.41\%)}$ | $1.4146_{(-1.50\%)}$ | $2.5740_{(-0.64\%)}$ | $87.6790_{(-4.24\%)}$ | $1.1039_{(-0.15\%)}$ |

Table 7: DA Accuracy and Forecast Skill results for DPO (Align-M), IPO, and DSPO.

| | MSE | MAE | WRMSE | | | |
| | | | u10 | v500 | z500 | t850 |
|---|---|---|---|---|---|---|
| *DA Accuracy* | | | | | | |
| ObsFree | 0.0643 | 0.1382 | 1.5275 | 2.7467 | 104.5531 | 1.0897 |
| Align-ObsFree-M | 0.0611 | **0.1348** | 1.4942 | 2.7202 | **96.8603** | 1.0866 |
| IPO-ObsFree-M | 0.0607 | 0.1355 | 1.4831 | 2.6944 | 101.7478 | **1.0807** |
| DSPO-ObsFree-M | **0.0604** | 0.1357 | **1.4780** | **2.6900** | 97.0645 | 1.0845 |
| Repaint | 0.0623 | 0.1368 | 1.5009 | 2.7027 | 103.3036 | 1.0911 |
| Align-Repaint-M | 0.0598 | 0.1344 | 1.4759 | 2.6836 | 96.8906 | 1.0889 |
| IPO-Repaint-M | 0.0597 | 0.1350 | 1.4682 | 2.6660 | 100.7377 | 1.0868 |
| DSPO-Repaint-M | **0.0590** | **0.1338** | **1.4528** | **2.6555** | **96.2449** | **1.0762** |
| DPS | 0.0609 | 0.1337 | 1.4441 | 2.5972 | 93.4654 | 1.1159 |
| Align-DPS-M | 0.0593 | 0.1317 | 1.4187 | 2.5800 | **88.4235** | 1.1100 |
| IPO-DPS-M | **0.0582** | **0.1315** | **1.4070** | **2.5539** | 89.4790 | **1.0977** |
| DSPO-DPS-M | 0.0585 | 0.1320 | 1.4102 | 2.5576 | 89.8927 | 1.1014 |
| *Forecast Skill* | | | | | | |
| ObsFree | 0.1146 | 0.1918 | 2.1182 | 4.1571 | 232.6243 | 1.4826 |
| Align-ObsFree-M | **0.1080** | **0.1872** | 2.0965 | 4.1091 | **225.7758** | **1.4636** |
| IPO-ObsFree-M | 0.1093 | 0.1882 | 2.0966 | 4.1162 | 227.8923 | 1.4661 |
| DSPO-ObsFree-M | 0.1086 | 0.1879 | **2.0838** | **4.1020** | 226.1687 | 1.4637 |
| Repaint | 0.1123 | 0.1898 | 2.0899 | 4.1030 | 228.2876 | 1.4672 |
| Align-Repaint-M | **0.1068** | **0.1860** | 2.0713 | 4.0665 | **222.9336** | 1.4516 |
| IPO-Repaint-M | 0.1083 | 0.1872 | 2.0712 | 4.0721 | 224.3243 | 1.4555 |
| DSPO-Repaint-M | 0.1080 | 0.1868 | **2.0663** | **4.0585** | 223.3857 | **1.4504** |
| DPS | 0.0969 | 0.1757 | 1.9230 | 3.7675 | 193.4167 | 1.3564 |
| Align-DPS-M | 0.0943 | 0.1734 | 1.9070 | 3.7287 | 189.6935 | 1.3428 |
| IPO-DPS-M | 0.0937 | **0.1732** | **1.9045** | 3.7275 | 189.5445 | **1.3412** |
| DSPO-DPS-M | **0.0932** | 0.1735 | 1.9062 | **3.7269** | **189.2176** | 1.3418 |

To demonstrate the flexibility of the Align-DA framework, we performed a preliminary experiment incorporating a second physical constraint: hydrostatic balance. Hydrostatic balance governs the vertical structure of the atmosphere, linking pressure and temperature profiles via the hydrostatic equation:

$$\frac{\partial \Phi}{\partial \ln P} = -RT, \tag{19}$$

where $P$ is pressure, $R$ is the specific gas constant, and $T$ is temperature. We quantify its imbalance using the metric $D_{\text{hydro}} = \left| \left( \frac{\partial \Phi}{\partial \ln P} + RT \right) \Big/ RT \right|$. Similar to the Equation 17, one can also define *Hydro-Score*.

This extends the multi-reward alignment from three preferences to four. In the table below, "-M" denotes the original three-preference alignment, while "-4" denotes the four-preference alignment

which includes the additional hydrostatic constraint. The results shown in Table 8 validate our framework's ability to easily leverage multiple, diverse physical constraints, often leading to further performance gains across various metrics.

Table 8: DA Accuracy and Forecast Skill with an additional hydrostatic balance reward ('-4').

| | MSE | MAE | WRMSE | | | |
| | | | u10 | v500 | z500 | t850 |
|---|---|---|---|---|---|---|
| *DA Accuracy* | | | | | | |
| ObsFree | 0.0643 | 0.1382 | 1.5275 | 2.7467 | 104.5531 | 1.0897 |
| Align-ObsFree-M | 0.0611 | 0.1348 | 1.4942 | 2.7202 | 96.8603 | 1.0866 |
| Align-ObsFree-4 | **0.0571** | **0.1335** | **1.4675** | **2.6546** | **94.3716** | **1.0649** |
| Repaint | 0.0623 | 0.1368 | 1.5009 | 2.7027 | 103.3036 | 1.0911 |
| Align-Repaint-M | 0.0598 | 0.1344 | 1.4759 | 2.6836 | 96.8906 | 1.0889 |
| Align-Repaint-4 | **0.0567** | **0.1333** | **1.4536** | **2.6327** | **95.7225** | **1.0756** |
| DPS | 0.0609 | 0.1337 | 1.4441 | 2.5972 | 93.4654 | 1.1159 |
| Align-DPS-M | 0.0593 | 0.1317 | 1.4187 | 2.5800 | 88.4235 | 1.1100 |
| Align-DPS-4 | **0.0588** | **0.1315** | **1.4155** | **2.5672** | **87.3151** | **1.0996** |
| *Forecast Skill* | | | | | | |
| ObsFree | 0.1146 | 0.1918 | 2.1182 | 4.1571 | 232.6243 | 1.4826 |
| Align-ObsFree-M | 0.1080 | 0.1872 | 2.0965 | 4.1091 | 225.7758 | 1.4636 |
| Align-ObsFree-4 | **0.1028** | **0.1841** | **2.0456** | **4.0024** | **219.8289** | **1.4423** |
| Repaint | 0.1123 | 0.1898 | 2.0899 | 4.1030 | 228.2876 | 1.4672 |
| Align-Repaint-M | 0.1068 | 0.1860 | 2.0713 | 4.0665 | 222.9336 | 1.4516 |
| Align-Repaint-4 | **0.1017** | **0.1831** | **2.0345** | **3.9911** | **217.5085** | **1.4304** |
| DPS | 0.0969 | 0.1757 | 1.9230 | 3.7675 | 193.4167 | 1.3564 |
| Align-DPS-M | 0.0943 | 0.1734 | 1.9070 | 3.7287 | 189.6935 | 1.3428 |
| Align-DPS-4 | **0.0938** | **0.1725** | **1.9059** | **3.7244** | **189.3845** | **1.3423** |

**Realistic observation.** We further evaluate the performance of AlignDA using real-world observational data by conducting experiments with the Global Data Assimilation System (GDAS) prepbufr dataset, which incorporates multi-source observational data. To ensure data quality, we apply a filtering process that eliminates observations exhibiting deviations exceeding 0.1 standard error when compared to the ERA5 reference data. Table 9 shows that when using real-world GDAS observations, Align-DPS-M still outperforms the standard DPS model. It achieves better accuracy across all measures, with notable improvements like a 3.89% decrease in MSE and a 2.82% decrease in WRMSE for z500. These results demonstrate that the alignment strategy is effective and beneficial for practical, real-world applications.

# G   Limitation and Future work

While our Align-DA demonstrates promising results, several aspects should be further studied. The current physical constraints, primarily geostrophic balance, may not fully capture the complexities of operational DA, suggesting a need for more comprehensive reward designs. Additionally, the framework relies on offline reinforcement learning, which is inherently sensitive to the quality and diversity of the pre-collected preference dataset. Incorporating online RL strategies could enable more adaptive policy updates and mitigate limitations imposed by static offline data. Moreover, the RL-based alignment framework introduced in this paper has potential beyond improving initial conditions; it can also be generalized to other stages of the weather forecasting workflow, such as forecast post-processing.

Table 9: Assimilation accuracy gains with GDAS observations. Percentage values quantify relative accuracy variants compared to the non-aligned baseline.

| | MSE | MAE | WRMSE | | | |
| | | | u10 | v500 | z500 | t850 |
|---|---|---|---|---|---|---|
| DPS | 0.0634 | 0.1365 | 1.3342 | 2.5642 | 99.2263 | 1.1343 |
| Align-DPS-M | 0.0610(−3.89%) | 0.1340(−1.87%) | 1.3200(−1.08%) | 2.5498(−0.57%) | 96.4988(−2.82%) | 1.1342(−0.01%) |

Figure 5: Visulaization of u700 at a 2019-01-26-18:00 UTC.

# H  More visualization

Here we provide more visualization results. In all figures in the appendix, 'ObsFree' means without observation integration baseline. 'Repaint' and 'DPS' represent repaint and diffusion posterior sampling guidance methods. Note that '-P' indicates single physical reward alignment, while '-M' signifies multi-reward alignment. First row: Reference fields showing ERA5 reanalysis (ground truth denoted as GT), background field, and observational data. Second row: Error reduction through single physical reward alignment, quantified as $|\boldsymbol{x}_a^{\text{ref}} - \boldsymbol{x}_{\text{GT}}| - |\boldsymbol{x}_a^{\text{Align-P}} - \boldsymbol{x}_{\text{GT}}|$. Third row: Error reduction through multi-reward alignment, quantified as $|\boldsymbol{x}_a^{\text{ref}} - \boldsymbol{x}_{\text{GT}}| - |\boldsymbol{x}_a^{\text{Align-M}} - \boldsymbol{x}_{\text{GT}}|$.

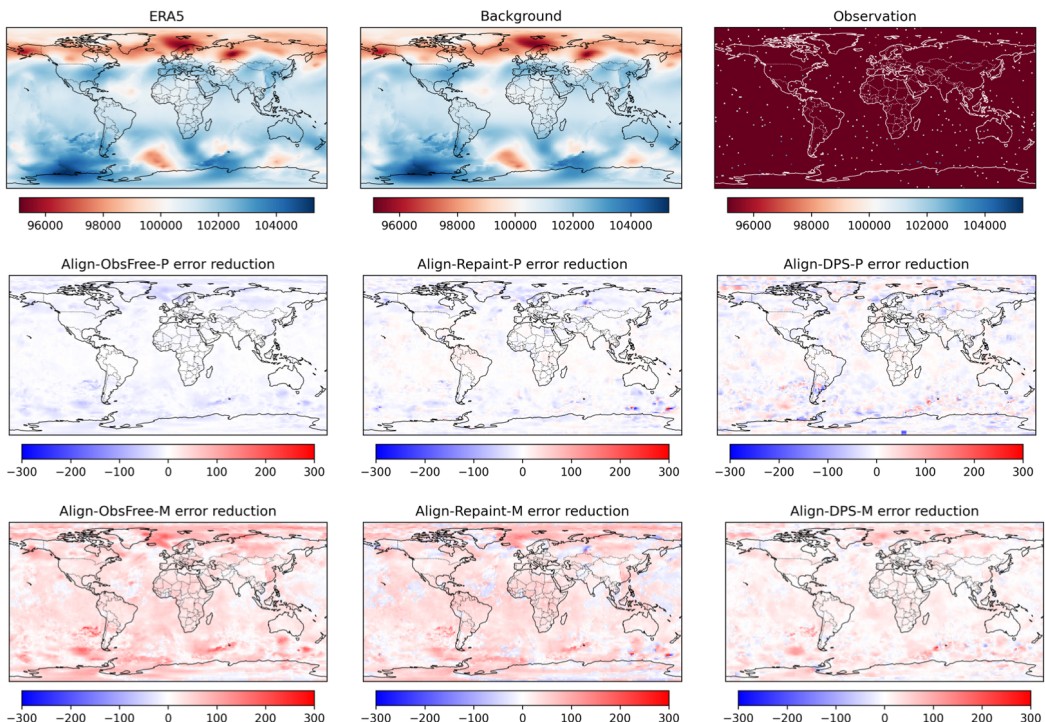

Figure 6: Visulaization of msl at a 2019-03-03-00:00 UTC.

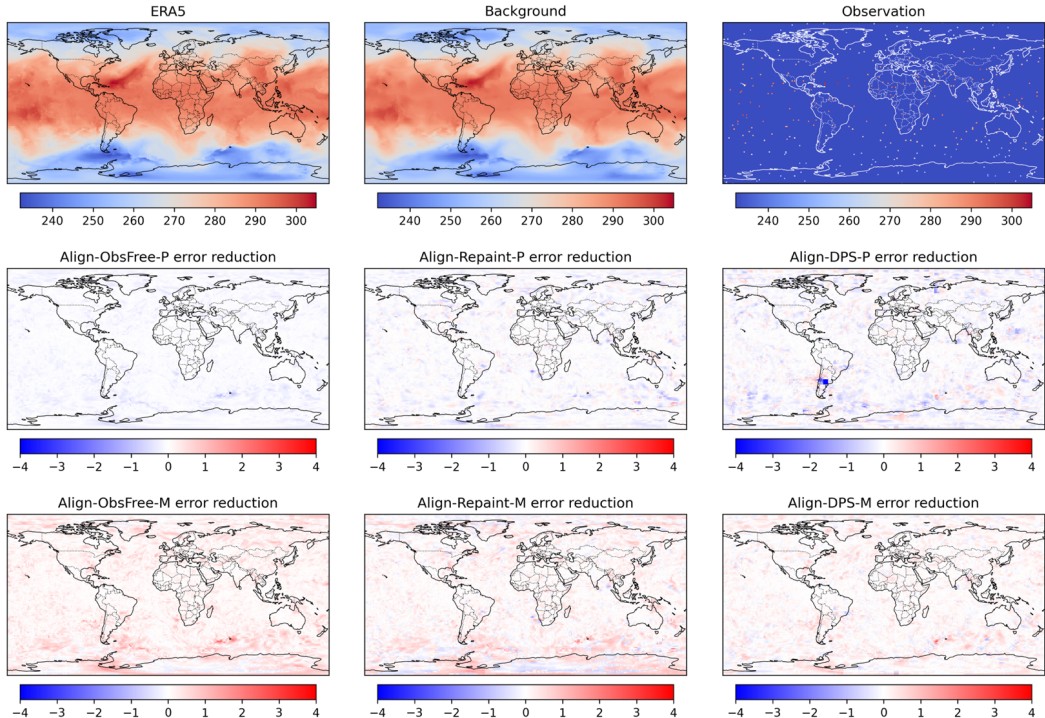

Figure 7: Visulaization of t850 at a 2019-07-05-12:00 UTC.

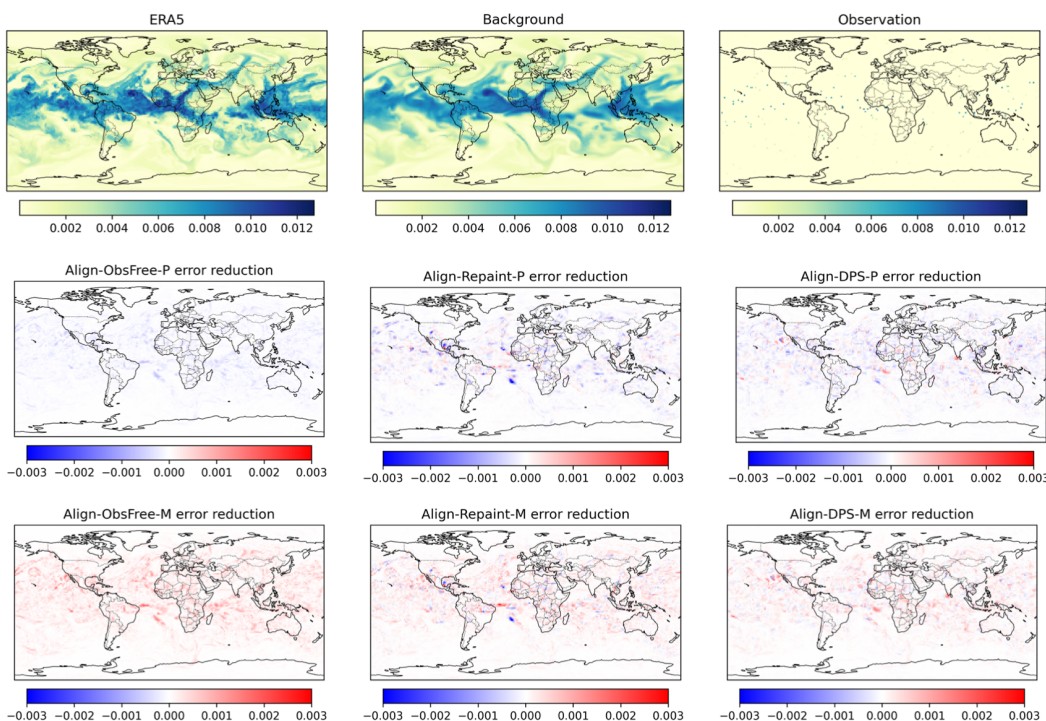

Figure 8: Visulaization of q700 at a 2019-04-01-00:00 UTC.

