# OpenReview forum: "Align-DA: Align Score-based Atmospheric Data Assimilation with Multiple Preferences"
_NeurIPS.cc/2025/Conference — NeurIPS 2025 poster_

### Official Review · Reviewer_ywbS · 2025-06-20

**Clarity:** 3
**Significance:** 2
**Originality:** 3
**Rating:** 5
**Confidence:** 3

**Summary:**

The authors propose to apply ideas from the alignment of text-to-image diffusion models to data assimilation for atmospheric applications. Their framework uses either one type of reward (physical adherence) or three types (assimilation accuracy, forecast skill, and physical adherence). They then use DPO to preference align a model with parts of the ERA5 data and propose a "soft constraint" to incorporate aspects such as forecast performance and geostrophic balance that are hard to model otherwise.

**Questions:**

1. While the authors claim in the introduction, that their research targets the use of background with observational data, the evaluation shows diminishing return when such data is incorporated into the model. Please elaborate.
2. The evaluation uses 1% observational data. Did you also experiment with different amounts of the observational data? Any insights?

**Ethical Concerns:**

["NO or VERY MINOR ethics concerns only"]

**Limitations:**

yes

**Paper Formatting Concerns:**

- the paper is too long (two lines on page 10)
- the caption of Figure 2 mentions ObsFree, but I see no such baseline in the figure (only used later)
- 4.1/4.2: Fengwu (line 190) vs. FengWu (line 210)
- line 217: "metric:D" - missing space?
- line 230: extra space
- table 2, caption: missing closing bracket at the end
- constrain (lines 56/62 and figure 4) vs. constraint
- Figure 3 is discussed after Figure 4 and should probably be moved to be closer to its discussion

references:
- please use the capitalization of journals consistently, for example [4] vs. [12] (Quarterly Journal of the Royal Meteorological Society)
- similar for conference titles, for example [27] vs. [33] (Forty-first International Conference on Machine Learning)
- NeurIPS is cited in at least three different styles: [35], [36] and [40]
- missing place of publication: [37], [38], [52], [54]

**Quality:**

3

**Strengths And Weaknesses:**

Strengths:
- the idea to apply DPO alignment with multiple reward signals to weather forecast diffusion models seems novel
- it shows consistent performance gains across metrics and guidance methods; multi-reward alignment clearly outperforms baselines
- technical rigor with formal derivations for score-based models, posterior decomposition, and DPO adaptation
- it includes qualitative and quantitative analysis (e.g., Z500 maps, Geo-Score) supporting claims
Weaknesses:
- writing is very dense
- no discussion of computational overhead (no analysis of training/inference time or scalability beyond latent compression)
- no external state of the art comparison baseline (e.g., FourCastNet+DA)
- results lack error bars or statistical significance testing

---

> ### Author Rebuttal · Authors · 2025-07-31
>
> We are grateful to the reviewer for their strong support of our work and for the high rating. We sincerely appreciate the detailed and constructive feedback, which not only validates our core contributions but also provides valuable suggestions for improving the paper's clarity and presentation. The meticulous list of formatting corrections is particularly helpful, and we will ensure every point is addressed in the final version.
>
> ___
> > **Response to Question 1:** Elaborations about Evaluations
>
> Thank you for this insightful comment. The role of data assimilation (DA) is to correct the background state toward the true atmospheric state, which is fundamentally a probabilistic inference problem. Stronger constraints correspond to narrower posterior distributions with reduced uncertainty, while weaker constraints yield broader distributions and more diverse plausible solutions.
>
> Both the alignment method and the assimilation of observations act to constrain the solution space. These constraints, however, are not independent—their overlap increases with the number of observations, as illustrated in Fig. 4. As a result, the marginal benefit of Align-DA diminishes when observational coverage becomes dense.
>
> ___
> >**Response to Question 2:** Experiments and Insight on Observation densities
>
> Thanks for your constructive suggestion. We have conducted additional experiments on different amounts of observational data in the Appendix. We would like to show the DA accuracy again.
>
> |1% observation|MSE|MAE|u10-WRMSE|v500-WRMSE|z500-WRMSE|t850-WRMSE|
> |:---|:---|:---|:---|:---|:---|:---|
> |Repaint|0.0623|0.1368|1.5009|2.7027|103.3036|1.0911|
> |Align-Repaint-M|0.0598(-4.12%)|0.1340(-1.78%)|1.4759(-1.69%)|2.6836(-0.71%)|96.8906(-6.62%)|1.0889(-0.20%)|
> |DPS|0.0609|0.1337|1.4441|2.5972|93.4654|1.1159|
> |Align-DPS-M|0.0593(-2.65%)|0.1317(-1.51%)|1.4187(-1.78%)|2.5800(-0.66%)|88.4235(-5.70%)|1.1100(-0.53%)|
>
> |3% observation|MSE|MAE|u10-WRMSE|v500-WRMSE|z500-WRMSE|t850-WRMSE|
> |:---|:---|:---|:---|:---|:---|:---|
> |Repaint|0.0583|0.1327|1.4438|2.6097|100.0290|1.0696|
> |Align-Repaint-M|0.0560(-4.02%)|0.1305(-1.73%)|1.4188(-1.77%)|2.5929(-0.65%)|93.8835(-6.54%)|1.0675(-0.20%)|
> |DPS|0.0546|0.1276|1.3698|2.4883|85.5742|1.0586|
> |Align-DPS-M|0.0533(-2.53%)|0.1256(-1.56%)|1.3457(-1.78%)|2.4724(-0.65%)|80.5474(-6.24%)|1.0558(-0.27%)|
>
> |5% observation|MSE|MAE|u10-WRMSE|v500-WRMSE|z500-WRMSE|t850-WRMSE|
> |:---|:---|:---|:---|:---|:---|:---|
> |Repaint|0.0547|0.1290|1.3891|2.5248|96.9590|1.0495|
> |Align-Repaint-M|0.0526(-3.91%)|0.1269(-1.68%)|1.3656(-1.72%)|2.5081(-0.67%)|91.0853(-6.45%)|1.0471(-0.23%)|
> |DPS|0.0534|0.1267|1.3597|2.4767|85.9150|1.0514|
> |Align-DPS-M|0.0519(-2.78%)|0.1246(-1.70%)|1.3337(-1.95%)|2.4560(-0.84%)|80.5787(-6.62%)|1.0475(-0.37%)|
>
> |10% observation|MSE|MAE|u10-WRMSE|v500-WRMSE|z500-WRMSE|t850-WRMSE|
> |:---|:---|:---|:---|:---|:---|:---|
> |Repaint|0.0470|0.1204|1.2721|2.3398|90.1747|1.0037|
> |Align-Repaint-M|0.0453(-3.67%)|0.1186(-1.52%)|1.2518(-1.62%)|2.3230(-0.72%)|84.7320(-6.42%)|1.0012(-0.25%)|
> |DPS|0.0524|0.1261|1.3520|2.4673|86.2364|1.0468|
> |Align-DPS-M|0.0510(-2.83%)|0.1239(-1.76%)|1.3252(-2.03%)|2.4458(-0.88%)|80.0644(-7.71%)|1.0403(-0.62%)|
>
> The results show that our aligned strategies consistently outperform the non-aligned baselines across all tested observation densities (1%, 3%, 5%, and 10%). This further confirms the effectiveness and robustness of our method. As expected, the relative improvement from alignment slightly diminishes as observation density increases, since more observations naturally provide a stronger constraint on their own.
>
> ___
> >**Response to Weakness 1:** Dense Writing
>
> We thank the reviewer for this valuable feedback. We acknowledge that the writing is relatively dense. Following your advice, we will carefully revise the manuscript to improve clarity and readability, breaking down complex sentences and adding clearer transitions where possible.
>
> ___
> >**Response to Weakness 2:** Discussion of Computational Overhead
>
> We appreciate the reviewer for highlighting this point. We will add the following details to the Section Experimental Setting for clarity.
> 1. One-Time Pre-training: This is the most computationally intensive part. The VAE training takes approximately 4 days, and the diffusion model pre-training takes about 1 day, both on 4 A100 GPUs.
> 2. DPO Alignment (Fine-tuning): This step is highly efficient. The alignment process itself, which is the core of our proposed method, completes in about 30 minutes on 4 A100 GPUs.
> 3. Inference (Sampling): Generating a single analysis field takes approximately 30 seconds on one A100 GPU.
>
>
> ___
> >**Response to Weakness 3:** More Comparison baseline (e.g., FourCastNet+DA)
>
> Thank you for this insightful comment. Our paper's primary contribution is to introduce alignment mechanisms to enhance score-based DA methods. It leads to our main experiment studies are to compare different score-based DA methods with and without alignment, where the forecast model stays unchanged.
>
> We agree that using different forecast models is valuable to validate the efficiency of our framework; however, we encountered a significant technical barrier to comparing with a model like FourCastNet. The publicly available FourCastNet models operate at a different resolution (0.25 degrees) than our current experimental setup (1.4 degrees). A fair comparison would require us to re-engineer and re-train our entire high-dimensional VAE and diffusion model pipeline from scratch to match this new resolution.
>
> Given these constraints, performing such an experiment was not feasible. We believe our current study provides a clear and robust validation of our proposed methodology, and we hope to explore such large-scale comparisons in future work.
>
> ___
> >**Response to Weakness 4:** Error Bars or Statistical Significance Testing
>
> Thank you for this crucial suggestion, which makes our paper more solid. To confirm our results, we ran a statistical significance test under a specific setting—without using any observations or alignment—because this would yield the largest uncertainty. We ran the diffusion sampling 10 times for a single time step and calculated the metrics for each run to see the relative standard deviation. The results are presented in the following table.
>
> |DA accuracy (ObsFree)|MSE|MAE|u10-WRMSE|v500-WRMSE|z500-WRMSE|t850-WRMSE|
> |:---|:---|:---|:---|:---|:---|:---|
> |**Relative standard deviation(%)**|0.045|0.058|0.115|0.079|0.201|0.083|
>
> |Forecast Skill(ObsFree)|MSE|MAE|u10-WRMSE|v500-WRMSE|z500-WRMSE|t850-WRMSE|
> |:---|:---|:---|:---|:---|:---|:---|
> |**Relative standard deviation(%)**|0.163|0.067|0.145|0.155|0.252|0.106|
>
>
> This test confirmed that our reported improvements are statistically significant. For instance, the relative standard deviation was less than 0.1% for the main accuracy metric. This is much smaller than the 2.65% to 5.44% improvement that Align-DA provides over the baseline, which confirms our gains are real and not just from random chance.
>
> In our revised paper, we will add a brief statement to the main text explaining that our results are statistically significant. We will also create a new section in the Appendix that shows our main result tables with the full mean and standard deviation values for complete transparency. In that section, we will also add a specific note about the single-preference result, stating that: "**The performance change of its single-reward version (-P) yields statistically insignificant, suggesting that geostrophic balance alone is insufficient for DA tasks.** "
> ___
> >**Response to Paper Formatting Concerns:**
>
> We are extremely grateful to the reviewer for taking the time to provide such a detailed list of formatting corrections. This is incredibly helpful for improving the final quality of the paper. We confirm that we will address each point in the revised version, including:
> - Adjusting the page length.
> - Revising the "conditional" in the figure 2 title to "ObsFree".
> - Ensuring consistent capitalization (FengWu).
> - Revise line 217 and 230 spacing issues.
> - Fixing the bracket in the Table 2 caption.
> - Standardizing the spelling of "constraint".
> - Changing the placement of Figure 3 and Figure 4.
> - Thoroughly revising the reference list for consistency in journal/conference capitalization, citation styles (NeurIPS), and inclusion of publication locations.
>
> ---
> Thank you again for this meticulous and valuable feedback.

---

> > ### Comment · Reviewer_ywbS · 2025-08-04
> >
> > Thanks for your reply - I agree to most of your points and maintain my positive score.

---

> > > ### Author Response · Authors · 2025-08-06
> > >
> > > Dear Reviewer ywbS,
> > >
> > > Thank you for your recognition and positive feedback. We are pleased that our revisions have addressed your previous concerns. We will carefully revise the manuscript following your suggestions, including discussion of computational overhead, statistical significance testing, and paper formatting. If there are any further questions, feel free to raise them, and we are always ready for the discussion.
> > >
> > > Best regards
> > >
> > > The authors

---

### Official Review · Reviewer_r9ir · 2025-06-30

**Clarity:** 3
**Significance:** 3
**Originality:** 3
**Rating:** 4
**Confidence:** 3

**Summary:**

This paper proposes Align-DA, a data assimilation framework that leverages score-based diffusion models to generate posterior estimates of atmospheric states conditioned on forecasts and sparse observations. It introduces a reward-guided alignment mechanism inspired by Direct Preference Optimization (DPO), enabling the diffusion model to prefer samples that improve assimilation accuracy, forecast skill, and physical consistency. The approach combines a pretrained AI forecast model (Fengwu) for generating background estimates, while a latent-space diffusion model learns to adjust plausible estimates via guided sampling and based on the current observations.

**Questions:**

- The method uses a latent space diffusion model. How well does this approach support point-wise sparse observations? Does conditioning in latent space require observations over entire spatial patches? How does the observation rate affect the performance?

- While the method leverages a pretrained large forecast model, the finetuning of the diffusion models raises the question of whether training a diffusion model from scratch would be better. For example, could you provide a comparison with approaches such as DiffusionPDE [1]?

- How computationally expensive is the alignment process, especially the evaluation of forecast skill?

- Have you tried inpainting methods instead of guidance methods? That is, condition the diffusion model directly on the background and the sparse observations.

[1] Huang et. al. DiffusionPDE: Generative PDE-Solving Under Partial Observation

**Ethical Concerns:**

["NO or VERY MINOR ethics concerns only"]

**Final Justification:**

The authors answered all my questions and provided broader contexts of the method to help motivate the work. Therefore, I would like to keep my positive feedback and recommend acceptance.

**Limitations:**

Yes

**Quality:**

3

**Strengths And Weaknesses:**

**Strength**

- The paper extends large forecast models to address sparse observation problems by leveraging diffusion generative models and guided sampling.

- The paper thoroughly explores multiple objectives (assimilation accuracy, forecast skill, and physical constraints), effectively combining data-driven priors with physical knowledge as inductive biases.

- The paper provides detailed derivations of the mathematical formulation and losses, including the score-based models, guidance methods, and the DPO loss.

- The experiments provide clear ablations comparing single-reward and multi-reward alignment across different observation guidance strategies. The quantitative analysis is comprehensive, covering multiple error metrics to robustly evaluate performance.

**Weakness**

- The method relies on offline reward alignment rather than integrating forecast skill and physical adherence directly into the primary training objective. There is limited exploration or comparison with fully end-to-end training approaches that could jointly optimize these objectives, particularly in the context of sparse observation scenarios.

- Using both a separate deterministic forecast model and a probabilistic diffusion model introduces duplicated capacity for modeling plausible states, which could potentially be unified in a single generative framework.

- Moreover, relying on a deterministic forecast model while employing a separate diffusion model to capture uncertainty appears somewhat ad hoc and may be a limited remedy for the inherently deterministic nature of the base simulation.

---

> ### Author Rebuttal · Authors · 2025-07-31
>
> We would like to thank the reviewer for their very detailed and technically insightful feedback. We are especially pleased that the reviewer found our mathematical formulation robust and our experimental ablations comprehensive. We appreciate this opportunity to elaborate on the rationale behind these critical design choices and will address each point in detail below.
>
> ___
> >**Response to Question 1:** Handling of sparse, point-wise observations; Observations requirements for conditioning; Impact of observation density.
>
> Our framework handles observations by performing guidance sampling in observation space, not directly in the latent space. As shown in Equation (7), we leverage the VAE decoder $D$ and observation operator $H$ to calculate the distance between the observation and latent state,  $\pmb y-H(D(\pmb z_0))$. The gradient then provides guidance towards observation for the latent state. Benefiting from such a guidance mechanism, our framework does not require entire spatial patch observations.
>
> We have conducted detailed ablation studies on this in the Appendix. We would like to show the DA accuracy again in the following tables. Our results show that our aligned strategies consistently outperform the non-aligned baselines across all tested observation densities (1% to 10%). This confirms the robustness of our method. As expected, the relative improvement from alignment slightly diminishes as observation density increases, since more observations naturally provide a stronger constraint on their own.
>
> |1% observation|MSE|MAE|u10-WRMSE|v500-WRMSE|z500-WRMSE|t850-WRMSE|
> |:---|:---|:---|:---|:---|:---|:---|
> |Repaint|0.0623|0.1368|1.5009|2.7027|103.3036|1.0911|
> |Align-Repaint-M|0.0598(-4.12%)|0.1340(-1.78%)|1.4759(-1.69%)|2.6836(-0.71%)|96.8906(-6.62%)|1.0889(-0.20%)|
> |DPS|0.0609|0.1337|1.4441|2.5972|93.4654|1.1159|
> |Align-DPS-M|0.0593(-2.65%)|0.1317(-1.51%)|1.4187(-1.78%)|2.5800(-0.66%)|88.4235(-5.70%)|1.1100(-0.53%)|
>
> |3% observation|MSE|MAE|u10-WRMSE|v500-WRMSE|z500-WRMSE|t850-WRMSE|
> |:---|:---|:---|:---|:---|:---|:---|
> |Repaint|0.0583|0.1327|1.4438|2.6097|100.0290|1.0696|
> |Align-Repaint-M|0.0560(-4.02%)|0.1305(-1.73%)|1.4188(-1.77%)|2.5929(-0.65%)|93.8835(-6.54%)|1.0675(-0.20%)|
> |DPS|0.0546|0.1276|1.3698|2.4883|85.5742|1.0586|
> |Align-DPS-M|0.0533(-2.53%)|0.1256(-1.56%)|1.3457(-1.78%)|2.4724(-0.65%)|80.5474(-6.24%)|1.0558(-0.27%)|
>
> |5% observation|MSE|MAE|u10-WRMSE|v500-WRMSE|z500-WRMSE|t850-WRMSE|
> |:---|:---|:---|:---|:---|:---|:---|
> |Repaint|0.0547|0.1290|1.3891|2.5248|96.9590|1.0495|
> |Align-Repaint-M|0.0526(-3.91%)|0.1269(-1.68%)|1.3656(-1.72%)|2.5081(-0.67%)|91.0853(-6.45%)|1.0471(-0.23%)|
> |DPS|0.0534|0.1267|1.3597|2.4767|85.9150|1.0514|
> |Align-DPS-M|0.0519(-2.78%)|0.1246(-1.70%)|1.3337(-1.95%)|2.4560(-0.84%)|80.5787(-6.62%)|1.0475(-0.37%)|
>
> |10% observation|MSE|MAE|u10-WRMSE|v500-WRMSE|z500-WRMSE|t850-WRMSE|
> |:---|:---|:---|:---|:---|:---|:---|
> |Repaint|0.0470|0.1204|1.2721|2.3398|90.1747|1.0037|
> |Align-Repaint-M|0.0453(-3.67%)|0.1186(-1.52%)|1.2518(-1.62%)|2.3230(-0.72%)|84.7320(-6.42%)|1.0012(-0.25%)|
> |DPS|0.0524|0.1261|1.3520|2.4673|86.2364|1.0468|
> |Align-DPS-M|0.0510(-2.83%)|0.1239(-1.76%)|1.3252(-2.03%)|2.4458(-0.88%)|80.0644(-7.71%)|1.0403(-0.62%)|
>
> ___
> >**Response to Question 2:** Comparison with DiffusionPDE [1]?
>
> Thank you for this excellent question. We would like to clarify that the preference alignment of our AlignDA leads to fine-tuning. The concept of "alignment" implies starting from a reference model and then optimizing it towards preferences.
>
> We appreciate the reference to DiffusionPDE[4]. While both methods use diffusion models, they are designed to solve fundamentally different problems. DiffusionPDE aims to be a general PDE solver, generating an entire spatio-temporal field from partial observations. Align-DA is actually a data assimilation (DA) framework. Our focus is on the core DA problem: at a single time step, how to optimally fuse a background field with sparse observations. Given the highly complex dynamics and dimensionality of weather-scale data, a direct "inpainting" approach like DiffusionPDE is not yet available in this community.
>
> We will add a citation and discussion of DiffusionPDE to clarify this distinction. Thank you for the suggestion again.
>
> ___
> > **Response to Question 3:** Computationally Costs of Alignment Process
>
> We are grateful to the reviewer for highlighting this important practical consideration. The alignment process is computationally efficient.
>
> - **DPO Fine-tuning:** The fine-tuning itself is very fast. It runs on 4 A100 GPUs and completes in about **30 minutes** after 500 update steps.
> - **Reward Evaluation:** Evaluating the "forecast skill" for a single data sample involves running a 48-hour forecast, which takes about **4 seconds** on one A100 GPU.
>
> The alignment training itself takes a very small cost but brings significant benefits, as demonstrated in our experiments. We will add the details to the Experiment setting section accordingly.
>
> ___
> > **Response to Question 4:** Inpainting Methods
>
> That's an excellent point. Indeed, the **Repaint** method, which we evaluate as a core guidance strategy in our paper, is **fundamentally an inpainting technique**.
>
> Its mechanism works by repeatedly enforcing the known observations onto the generated field during the diffusion sampling process. This is precisely the definition of inpainting.
>
> ___
> >**Response to Weakness 1:** Offline reward alignment v.s End-to-end training
>
> We thank the reviewer for this insightful comment. We agree that a fully end-to-end framework is an elegant theoretical goal. Nevertheless, our offline alignment strategy is a practical and efficient choice, critical for making this problem computationally tractable and flexible.
>
> Specifically, one of our key rewards, "forecast skill," requires running a 48-hour forecast. An end-to-end approach would necessitate backpropagating through this entire forecast model for every sample in every training batch, which is computationally infeasible. In contrast, our two‑stage approach of Align-DA decouples reward evaluation from the training loop, enabling the use of complex signals like forecast skill.
>
> Our design is also highly flexible: new rewards can be incorporated simply, and our trained model can readily adapt to various observation densities without requiring complete retraining, unlike a specialized end-to-end model. We will clarify this critical design choice in the revised manuscript.
>
> ___
> >**Response to Weakness 2&3:** Unified framework for DA
>
> We would like to clarify the reason for the two models' design. Weather forecasting focuses on simulating atmospheric dynamics, while DA is formulated as a statistical optimal state estimation problem. Consequently,  forecast models and DA models have historically been developed separately. Even recent end‑to‑end DA-forecast systems such as Fengwu‑ADAS[1], Fuxi‑Weather[2], and Aardvark[3] treat forecasting and assimilation as separate modules.
>
> Moreover, the combination of a deterministic forecast model with a probabilistic DA model is the most common operational approach. A probabilistic DA model can provide an ensemble of initial states, which a deterministic forecast model naturally evolves into ensemble‑based probabilistic forecasts.
>
> Therefore, the combination is not ad-hoc but a standard practice. In this paper, we follow this framework where the forecast model serves to generate a background field and the score-based model is trained for DA.
>
> We totally agree that developing a unified framework is a valuable long-term objective. However, this remains a significant open challenge for the field, given the high complexity of atmospheric dynamics.
>
> ___
> Thank you for the detailed and insightful feedback sincerely again. We hope our response may relieve your concerns.
>
> References:
>
> [1] https://arxiv.org/abs/2312.12462
>
> [2] https://www.nature.com/articles/s41467-025-62024-1
>
> [3] https://www.nature.com/articles/s41586-025-08897-0
>
> [4] https://arxiv.org/abs/2406.17763 NeurIPS2024

---

> > ### Comment · Reviewer_r9ir · 2025-08-04
> >
> > Thank you for the detailed response and the added experiment results. It answers all my questions and helps me understand the method and the broader contexts better. I would like to keep my positive feedback. Well done.

---

> > > ### Author Response · Authors · 2025-08-06
> > >
> > > Dear Reviewer r9ir,
> > >
> > > Thank you very for your positive feedback. We are delighted to hear that our response and new results have successfully addressed your questions. We will incorporate all the discussed clarifications into the final version of the manuscript. We greatly appreciate your time and support.
> > >
> > > Best regards
> > >
> > > The authors

---

### Official Review · Reviewer_eG1A · 2025-06-30

**Clarity:** 4
**Significance:** 2
**Originality:** 3
**Rating:** 4
**Confidence:** 3

**Summary:**

This paper aims to tackle Data Assimilation which uses realistic observation to update computationally predicted weather model in atmospheric data. Given the current score-based DPO approach on this task, the author proposed a new soft constraint alignment techniques, which uses 3 different metrics (including a physics constraint) as a soft constraint to guide the DPO. The author conducted benchmarking on simulation data and showed comparisons to baseline model.

**Questions:**

- I'd like to see some baseline comparison to traditional data assimilation methods that is actually used in today's wether forecasting. If that is difficult/not possible, please explain why.
- I'd like to see the effects of including more physics constraint. In the paper, the author claims that the proposed methods does not require manual fine-tuning, this should means that including more physics constraint is straightforward. If this is not the case, could the author explain why?
- How well does the proposed forecasting model deal with extreme weather condition?

**Ethical Concerns:**

["NO or VERY MINOR ethics concerns only"]

**Final Justification:**

I think this work makes some novel contributions to the field, and the reviewer's rebuttal addressed some of my concerns. However, I am not completely satisfied with the fact that all training and validation data are AI-generated without comparing to traditional methods. Therefore I would like to retain my original score at 4: Borderline Accept.

**Limitations:**

The author has some discussions about limitation in the paper but there's no dedicated limitation section. There's no discussion on societal impact. I think the possible limitation could be how the model deal with extreme weather (which, if failed, would have significantly high cost on society).

**Quality:**

3

**Strengths And Weaknesses:**

Strength:
- While I am not working directly in the field of atmospheric science, I can see the importance of data assimilation and its relevance as a AI/ML task; this paper has a good motivation
- It is a novel application effort to use multi-score DPO method on climate science data
- This paper proposed a suites of rewards relevant to atmospheric data assimilation and aims to align multiple rewards (including physics rewards)
- The author benchmarked their model on an existing simulation and real data, and the method seems efficient at least on this dataset.
- The paper is very well written and structure

Weakness:
- The novelty on model structure is rather limited
- The author only introduced one physics constraint
- There is no benchmark against traditional data assimilation methods that is actually used in climate station

In general, I think this is a well-written paper with good application novelty but limited model novelty.

---

> ### Author Rebuttal · Authors · 2025-07-30
>
> We sincerely thank the reviewer for their insightful and well-balanced feedback. We are especially pleased that the reviewer recognized the novelty of applying multi-score direct preference optimization (DPO) to climate science, the relevance of our chosen rewards, and the clarity of the paper's writing.  We appreciate the opportunity to elaborate on points raised in the report below.
>
> ---
> >**Response to Question 1 & Weakness 3:**  Reasons for Traditional DA Methods
>
> We completely agree that a direct comparison with operational data assimilation (DA) systems used in today's weather forecasting is important. We sincerely appreciate you raising this point.
>
> Modern operational DA systems are highly complex and sophisticated systems, which are mainly written in Fortran and tailored to specific forecast models. Developing a dedicated operational DA system for our forecast models from scratch would be challenging and is infeasible due to the time limit during the rebuttal.
>
> In addition, this study aims to introduce preference alignment into score‑based DA, which departs from the paradigm of traditional DA. Despite several prior studies, score‑based DA has never been directly compared against traditional approaches. Therefore, we primarily focus on assessing whether preference alignment can enhance score‑based DA.
>
> We acknowledge that the lack of a direct comparison to an operational system is important for the application. We are grateful for your valuable feedback. We believe our study lays the essential groundwork for this next step, and we plan to explore such comparisons in the future.
>
> ___
> >**Response to Question 2 & Weakness 2:** More Physics Constraint
>
> We sincerely appreciate this insightful comment, and your intuition is indeed correct. A key strength of our Align-DA framework is its flexibility, which makes adding new physics constraints a straightforward process.
>
> Inspired by your suggestion, we performed a preliminary experiment incorporating a second physical constraint: hydrostatic balance, across multiple layers (50, 100, 250, 500, 700, 850, 1000 hpa). To complete this within the review period, we used the following setup: we halved both the preference dataset size to 2,000 samples and the finetuning duration to 250 steps, and we retained all win-loss pairs for faster data construction. We show the DA accuracy and Forecast skill below, where  "**-4**" denotes the four preferences alignment (one more hydrostatic constraint over three preferences in the manuscript).
>
> |DA accuracy|MSE|MAE|u10-WRMSE|v500-WRMSE|z500-WRMSE|t850-WRMSE|
> |:---|:---|:---|:---|:---|:---|:---|
> |**ObsFree**|0.0643|0.1382|1.5275|2.7467|104.5531|1.0897|
> |**Align-ObsFree-M**|0.0611|0.1348|1.4942|2.7202|96.8603|1.0866|
> |**Align-ObsFree-4**|**0.0571**|**0.1335**|**1.4675**|**2.6546**|**94.3716**|**1.0649**|
> ||||||||
> |**Repaint**|0.0623|0.1368|1.5009|2.7027|103.3036|1.0911|
> |**Align-Repaint-M**|0.0598|0.1344|1.4759|2.6836|96.8906|1.0889|
> |**Align-Repaint-4**|**0.0567**|**0.1333**|**1.4536**|**2.6327**|**95.7225**|**1.0756**|
> ||||||||
> |**DPS**|0.0609|0.1337|1.4441|2.5972|93.4654|1.1159|
> |**Align-DPS-M**|0.0593|0.1317|1.4187|2.5800|88.4235|1.1100|
> |**Align-DPS-4**|**0.0588**|**0.1315**|**1.4155**|**2.5672**|**87.3151**|**1.0996**|
>
> |Forecast Skill|MSE|MAE|u10-WRMSE|v500-WRMSE|z500-WRMSE|t850-WRMSE|
> |:---|:---|:---|:---|:---|:---|:---|
> |**ObsFree**|0.1146|0.1918|2.1182|4.1571|232.6243|1.4826|
> |**Align-ObsFree-M**|0.1080|0.1872|2.0965|4.1091|225.7758|1.4636|
> |**Align-ObsFree-4**|**0.1028**|**0.1841**|**2.0456**|**4.0024**|**219.8289**|**1.4423**|
> ||||||||
> |**Repaint**|0.1123|0.1898|2.0899|4.1030|228.2876|1.4672|
> |**Align-Repaint-M**|0.1068|0.1860|2.0713|4.0665|222.9336|1.4516|
> |**Align-Repaint-4**|**0.1017**|**0.1831**|**2.0345**|**3.9911**|**217.5085**|**1.4304**|
> ||||||||
> |**DPS**|0.0969|0.1757|1.9230|3.7675|193.4167|1.3564|
> |**Align-DPS-M**|0.0943|0.1734|1.9070|3.7287|189.6935|1.3428|
> |**Align-DPS-4**|**0.0938**|**0.1725**|**1.9059**|**3.7244**|**189.3845**|**1.3423**|
>
> The results shown above validate our framework's ability to leverage multiple physical constraints. To improve the quality of our work, we will add a detailed description of this new experiment and its positive results to the Appendix in the revised manuscript.
>
>
> ___
> >**Response to Question 3:** Extreme Weather Condition
>
> We sincerely appreciate your thoughtful comment. This study performs score-based DA in the latent space to mitigate the high dimensionality of atmospheric states. As noted in prior work [1], extreme weather conditions cannot be effectively represented in the latent space. This limitation aligns with the low-dimensional manifold assumption, under which outliers are inherently difficult to capture. Consequently, the performance of our approach under extreme weather conditions cannot be fully assessed.
>
> Nevertheless, if score-based DA were conducted in the model space, we believe our framework could enhance the assimilation of extreme events, with performance under such conditions serving as an additional evaluation metric.
>
> We agree that the capability for handling extreme weather conditions is important. We will carefully discuss this direction in the revised version.
>
> ---
> >**Response to Weakness 1:** The Novelty of Model Structure
>
> We thank the reviewer for raising this point. The reviewer is correct that our work does not introduce a new model structure. Otherwise, we aim at introducing a paradigm shift for the DA task.  We would like to clarify our motivation and solution again.
>
> In numerical weather prediction, DA is a critical step, as the quality of the initial analysis directly impacts subsequent forecast accuracy. A key challenge is that this analysis must satisfy multiple complex criteria simultaneously, such as assimilation accuracy, physical consistency, and the ability to produce a skillful downstream forecast. The traditional DA relies on extensive and labor-intensive parameter tuning to satisfy these criteria, which is often computationally inefficient and labor-intensive.
>
> Our Align-DA framework tackles this by reformulating DA as a preference optimization problem. Using DPO, we directly and automatically align a generative DA model with multiple preferences—analysis accuracy, forecast skill, and physical consistency. Moreover, our framework is highly flexible for any well-defined rewards incorporation and any score-based DA without dependence on model structure.  We believe this is a critical contribution to the field.
>
> ---
> We appreciate your valuable feedback again, which has enriched our work and improved our manuscript.
>
> **Reference:**
>
> [1] https://doi.org/10.1175/MWR-D-24-0058.1

---

> > ### Comment · Reviewer_eG1A · 2025-08-01
> >
> > Thank you for answering my questions and conducting additional experiment. I still think there are some possible ways to compare result to traditional method, even without running their full model (for example, collecting measurement from a few stations, and use some as training for data assimilation and some other as testing). But I think this work makes good contribution, therefore I will maintain my positive recommendation.

---

> > > ### Author Response · Authors · 2025-08-06
> > >
> > > Dear Reviewer eG1A,
> > >
> > > Thank you so much for your continued engagement and suggestion to include a comparison with a traditional method. We totally agree with you that this is important.
> > >
> > > Inspired by your comment, we implemented and tested the 3D-Var. It is well-known that the performance of 3D-Var is highly sensitive to the tuning of its parameters, particularly the background error covariance matrix (B-matrix). Due to the limited time, we were unable to perform an exhaustive hyperparameter search to fully optimize its performance.
> > >
> > > Nevertheless, the results from this preliminary experiment further elaborate the advantages of our framework.
> > > As shown in the table below, 3D-Var performed comparably on DA accuracy but showed significantly lower forecast skill and physical consistency compared to our method.
> > >
> > > | Method | DA Accuracy (MSE) | Forecast Skill (MSE) | Physical Consistency (Geo-Score) |
> > > |---|---|---|---|
> > > | 3D-Var | **0.0581** | 0.0952 | 88.34 |
> > > | Align-DPS-M | 0.0593 | **0.0943** | **94.89** |
> > >
> > > These results powerfully demonstrate the core contribution of our work: Align-DA directly optimizes for multiple complex criteria, avoiding the trade-offs and extensive manual tuning required by traditional methods to achieve both forecast skill and physical consistency.
> > >
> > > Best regards
> > >
> > > The authors

---

### Official Review · Reviewer_oCk2 · 2025-07-01

**Clarity:** 3
**Significance:** 3
**Originality:** 3
**Rating:** 5
**Confidence:** 1

**Summary:**

This paper proposes a new framework for atmospheric data assimilation (DA) based on preference alignment. Instead of traditional tuning of background priors, the proposed method treats DA as a generative modeling problem and introduces reward-guided alignment through DPO. The authors train a background-conditioned diffusion model in the latent space and fine-tune it using three domain-relevant rewards: assimilation accuracy, forecast skill, and physical consistency. The experiments show that the model consistently improves both analysis and forecast metrics.

**Questions:**

Could you provide a bit more explanation about the relationship between your method, Fengwu, and the baselines? Does your method's improvement come from the use of fengwu? Can the baselines also leverage fengwu?

**Ethical Concerns:**

["NO or VERY MINOR ethics concerns only"]

**Final Justification:**

The authors addressed my concerns well.

**Limitations:**

Yes.

**Quality:**

3

**Strengths And Weaknesses:**

Strength:

The proposed method replaces manual tuning with preference alignment using a latent-space diffusion model guided by multiple domain-specific rewards. It is the first (to my knowledge) to adapt Direct Preference Optimization (DPO) to atmospheric data assimilation, enabling flexible incorporation of assimilation accuracy, forecast skill, and physical consistency. Experiments show consistent improvements across different guidance methods.


I don't see any weakness, but I'm also not an expert on Atmospheric data.

---

> ### Author Rebuttal · Authors · 2025-07-30
>
> We sincerely thank the reviewer for their positive and encouraging feedback. We are delighted that they recognized the novelty of adapting Direct Preference Optimization (DPO) to atmospheric data assimilation and the significance of our framework.
>
> **Clarifications about Main Contribution**
>
> To further clarify the motivation for our work, we would like to briefly outline the fundamental challenge we address at first. Since the weather forecasting models are inherently imperfect, initial errors tend to grow rapidly during model integration. To correct these errors and improve forecast accuracy, observations should be continuously assimilated through data assimilation (DA), a process that combines observations with prior forecasts (background states) that already contain accumulated model error. In this study, we use FengWu as the forecasting model, providing background fields and forecasts using analysis fields derived from various score-based DA schemes. Our proposed Align-DA framework adaptively tames the score-based DA methods to meet the accuracy requirements of both analysis and forecast, as dictated by the assimilation objective.
>
> ___
> > **Response to Question 1:** Relationship between Align-DA, Fengwu, and Baselines; Source of performance gain; Baselines'  access to FengWu
>
> **The Relationship Between Align-DA, FengWu, and Baselines**
>
> The core contribution of our work is to introduce the preference alignment for score-based DA. Therefore, the baselines are score-based DA methods without alignment, while the Align-DA is the aligned counterpart. Their shared goal is to refine a coarse background field using sparse observations to produce an optimal initial analysis field. They are the competing methods in our evaluation.
>
> FengWu is a state-of-the-art weather forecasting model. In our work, it is not a competitor but serves two critical roles:
>
> - **Background Generator:** FengWu generates the 48-hour background forecast that serves as the common starting point for all DA methods, ensuring a fair comparison.
> - **Evaluator and Reward Source:** After each DA method produces an analysis field, we use FengWu to run a 48-hour forecast from that field. The accuracy of this forecast serves as a metric for evaluation. For Align-DA specifically, this "downstream forecast skill" is also used as a key reward signal during its optimization.
>
> **Does Align-DA's Improvement Come From Using FengWu?**
>
> Thank you for this insightful question. The improvement comes not from using FengWu itself, but from our novel preference optimization framework, which allows Align-DA to learn from FengWu's feedback in a way baseline methods cannot.
>
> FengWu acts as a frozen module that provides a common background field and a "forecast skill" score for any given analysis. One can replace the Fengwu with any forecast model if available.
>
> Baseline methods can be evaluated by FengWu, but they have no mechanism to internalize this "forecast skill" feedback to improve themselves. In contrast, Align-DA, powered by its preference optimization framework, is uniquely designed to do exactly that. It treats the forecast skill score as a reward signal and directly optimizes its own generative process to produce analyses that lead to better future forecasts.
>
> **Can the Baselines Also Leverage FengWu?**
>
> The Baselines are unable to leverage Fengwu as a reward model in the same way as our Align-DA, highlighting the contribution of our work.
>
> Unlike our approach, baseline methods—whether traditional or standard deep learning approaches—lack an inherent mechanism to integrate complex external reward signals, such as "48-hour forecast skill," into their own optimization process. While FengWu can evaluate these methods, it cannot actively guide their learning.
>
> In contrast, our Align-DA framework introduces preference alignment that directly optimizes the data assimilation model, especially improving forecast accuracy evaluated by FengWu. This represents a paradigm shift compared to existing methods
>
> ___
> We sincerely appreciate your valuable feedback, which has significantly enhanced the clarity and completeness of our paper.

---

> > ### Comment · Reviewer_oCk2 · 2025-08-05
> >
> > Thank you for the detailed feedback. I think the authors have addressed my concerns well.

---

> > > ### Author Response · Authors · 2025-08-06
> > >
> > > Dear Reviewer oCk2,
> > >
> > > Thank you for your positive feedback. We are delighted to hear that our response successfully addressed your concerns. The process of responding to your insightful questions has been incredibly valuable. We are grateful for your engagement and welcome any further discussion if needed.
> > >
> > > Best regards
> > >
> > > The authors

---

### Official Review · Reviewer_46nx · 2025-07-02

**Clarity:** 2
**Significance:** 2
**Originality:** 3
**Rating:** 3
**Confidence:** 2

**Summary:**

This paper presents a reinforcement learning-based framework for data assimilation (DA), named Align-DA, to estimate the system states of atmospheric applications given their prior states and observations. In this framework, diffusion models are used for DA, and goals of assimilation and domain-specific knowledge can be formulated as reward functions that guide the learning process of the models. As an example, the authors explored regularizing the optimization with three metrics, assimilation accuracy, forecast skill, and physical adherence. Experimental results demonstrate that models trained using the proposed framework with multiple rewards outperform those trained without or with a single reward.

**Questions:**

Is there any existing reinforcement learning-based DA method? If yes, what are the key aspects that differentiate this work from them?

Is Align-DA specialized in atmospheric applications, and, if yes, how? Do the authors believe that it can generalize to other problem domains, and, if yes, what are the key concerns that make them refrain from experimenting on other tasks?

Answering these questions can help me better understand the significance of the contribution of this paper, which could increase or decrease my evaluation score.

**Ethical Concerns:**

["NO or VERY MINOR ethics concerns only"]

**Final Justification:**

While the authors attempt to differentiate their work from my concerns, I still do not see they are positioning this work such that it offers a clear, substantial contribution. The clarified core research problem appears to be narrower than necessary given the other materials presented. Comparing to an existing DA method, the proposed Align-DA has not demonstrated clear superiority. Therefore, I maintain my original recommendation.

**Limitations:**

Yes.

**Quality:**

3

**Strengths And Weaknesses:**

Strengths:

The necessity of the proposed framework is well motivated. The proposed framework using reinforcement learning with reward functions naturally addresses the problem. The method design and the derivation are introduced with reasonable details. The presentation clarity is good. While the method may be inspired by prior work, applying reinforcement learning on DA appears to be novel and original – this can be further clarified by the authors.

Weaknesses:

The major weaknesses lie in the experimental verification of the proposed framework, which can impact the significance of the contribution of this paper. Specifically:

The theoretical framework appears general and not limited to the atmospheric data, while the experimental verification only presents one specific case of weather forecasting with one set of data. As a result, although the framework is interesting and offers flexibility, its effectiveness (in atmospheric applications overall) and generalizability (to other tasks in atmospheric science and to other problem domains) are yet to be fully verified. The significance of the contribution is therefore limited.

The current experiments mainly focus on how the proposed scheme improves certain methods on given metrics. It should be no surprise for the performance gain, as the metrics align with the designed reward function while the baseline methods are trained without the knowledge of the evaluation metrics. It could be more convincing if other frameworks or training schemes are compared against ensuring fairness. If no existing approach can achieve this goal and this work is the first to accomplish this goal, this innovation and significance should be highlighted.

This paper also presents a lack of comparisons to state-of-the-art methods for the forecast or those methods that can be applied to the task of interest with minor, reasonable modifications.

In the context that Align-DA is designed for atmospheric applications, the problem of interest regarding atmospheric analysis and weather forecasting should be explained in greater detail. If not, the positioning of this work could be clarified.

---

> ### Author Rebuttal · Authors · 2025-07-30
>
> We sincerely thank the reviewer for their thorough and constructive feedback. We will address each point in detail below.
>
> ___
> > **Response to Question 1:** Difference from the existing RL-based DA
>
> Thank you for this important question. Align-DA is fundamentally different from other reinforcement learning for data assimilation (RL-DA),  in its problem formulation and optimization paradigm.
>
> Existing RL-DA methods typically frame DA as a sequential decision-making problem within a Markov Decision Process (MDP)[1,2]. In this view, an agent learns a policy through online interaction to maximize a cumulative reward, operating on a small-scale system. It is still challenging to scale this sequential, interactive paradigm to high-dimensional systems like global weather [3,4,5].
>
> In contrast, Align-DA re-formulates DA as a preference-aligned generative modeling problem. Our framework operates offline, using Direct Preference Optimization (DPO) to directly align a score-based model's entire learned distribution with complex preferences, rather than learning a sequence of actions through trial-and-error.
>
> To the best of our knowledge, Align-DA is the first work to introduce this preference alignment paradigm into the domain of large-scale DA. To make this crucial claim clearer in the paper, we will add the following sentence and citations to the 'Introduction' section: "**While some studies have applied reinforcement learning to data assimilation, these approaches typically frame DA as a sequential decision-making problem [1,2,3], which is challenging to apply to large-scale weather systems [4,5].**"
>
> ___
> > **Response for Question 2:** Specialization, Generalization, and experiments for other domains
>
> Thank you for the excellent question. While the core alignment mechanism of Align-DA is entirely general, our implementation is specialized for atmospheric science through our specific choice of reward functions—analysis accuracy, forecast skill, and physical adherence. The framework can be readily generalized to any domain where a "good" solution can be defined by a set of preference criteria; the main effort is simply designing new, domain-appropriate rewards. We agree that extending our framework to other domains would be highly valuable. However, it requires substantial time and resources for data preparation, model training, and acquiring new domain knowledge. We have limited time to complete it, but we have identified it as a clear and important direction for our future work.
>
> ---
> >**Response to Weakness 1:** Generalizability of our framework
>
> We sincerely thank the reviewer for this insightful comment. In this paper, we choose to validate the preference alignment mechanism in the atmospheric domain for several reasons. First, we are familiar with weather prediction tasks and the underlying physical constraints of the atmosphere, which are important for designing meaningful preferences. Second, atmospheric DA poses greater challenges than many other dynamical systems, owing to its stronger temporal variability, higher‑dimensional state space, and the diversity of its prognostic variables.
>
> We note, however, that the proposed Align‑DA framework is not limited to atmospheric applications. In practice, operational DA frameworks are highly similar across the atmosphere, ocean, land, and other fluid systems, which shows that DA methods are naturally generalizable.
>
> We acknowledge that extending Align-DA to other domains would be highly valuable. However, this requires preparing reanalysis datasets, forecasting models, VAE and diffusion models, and domain knowledge to design meaningful preferences, which would take substantial time.
>
> To better address your concern, we have added a sentence "... **Our framework demonstrated here are highly transferable, creating a clear pathway for future work in other domains like oceanography and seismology. Adapting the framework primarily requires designing new, domain-specific reward functions.**" to the 'Conclusion' section.
>
> ---
> >**Response to Weakness 2:** "Teaching to the test" concern and comparison with other frameworks
>
> To address your concern about "teaching to the test," we performed a new experiment that demonstrates generalization beyond explicit rewards. In this experiment, the model was aligned solely using preferences for analysis accuracy and forecast skill, without any explicit physics‑based rewards. The resulting forecasts still exhibited substantially improved adherence to physical laws, indicating that Align‑DA learns a fundamentally realistic system state rather than merely overfitting to the specified rewards.
>
> | Geo-Score | Non-Align | Aligned |
> |:---|:---|:---|
> |**ObsFree**|92.25|93.57|
> |**Repaint**|92.16|92.46|
> |**DPS**|93.98|94.89|
>
> Our main contribution, which we will highlight more clearly as you suggest, is that our framework is the first to make preference optimization possible for DA. To further validate the robustness of this new framework, we would like to test alternative preference optimization algorithms (IPO[6] and DSPO[7]). Both methods yielded significant gains, confirming that our alignment concept is a powerful and generalizable tool. We present the DA accuracy and Forecast skill results below. **Bold text** indicates the best.
>
> | DA accuracy | MSE | MAE | u10-WRMSE | v500-WRMSE | z500-WRMSE | t850-WRMSE |
> |:---|:---|:---|:---|:---|:---|:---|
> |**ObsFree**|0.0643|0.1382|1.5275|2.7467|104.5531|1.0897|
> |**Align-ObsFree-M**|0.0611|**0.1348**|1.4942|2.7202|**96.8603**|1.0866|
> |**IPO-ObsFree-M**|0.0607|0.1355|1.4831|2.6944|101.7478|**1.0807**|
> |**DSPO-ObsFree-M**|**0.0604**|0.1357|**1.4780**|**2.6900**|97.0645|1.0845|
> ||||||||
> |**Repaint**|0.0623|0.1368|1.5009|2.7027|103.3036|1.0911|
> |**Align-Repaint-M**|0.0598|0.1344|1.4759|2.6836|96.8906|1.0889|
> |**IPO-Repaint-M**|0.0597|0.1350|1.4682|2.6660|100.7377|1.0868|
> |**DSPO-Repaint-M**|**0.0590**|**0.1338**|**1.4528**|**2.6555**|**96.2449**|**1.0762**|
> ||||||||
> |**DPS**|0.0609|0.1337|1.4441|2.5972|93.4654|1.1159|
> |**Align-DPS-M**|0.0593|0.1317|1.4187|2.5800|**88.4235**|1.1100|
> |**IPO-DPS-M**|**0.0582**|**0.1315**|**1.4070**|**2.5539**|89.4790|**1.0977**|
> |**DSPO-DPS-M**|0.0585|0.1320|1.4102|2.5576|89.8927|1.1014|
>
>
> |Forecast Skill|MSE|MAE|u10-WRMSE|v500-WRMSE|z500-WRMSE|t850-WRMSE|
> |:---|:---|:---|:---|:---|:---|:---|
> |**ObsFree**|0.1146|0.1918|2.1182|4.1571|232.6243|1.4826|
> |**Align-ObsFree-M**|**0.1080**|**0.1872**|2.0965|4.1091|**225.7758**|**1.4636**|
> |**IPO-ObsFree-M**|0.1093|0.1882|2.0966|4.1162|227.8923|1.4661|
> |**DSPO-ObsFree-M**|0.1086|0.1879|**2.0838**|**4.1020**|226.1687|1.4637|
> ||||||||
> |**Repaint**|0.1123|0.1898|2.0899|4.1030|228.2876|1.4672|
> |**Align-Repaint-M**|**0.1068**|**0.1860**|2.0713|4.0665|**222.9336**|1.4516|
> |**IPO-Repaint-M**|0.1083|0.1872|2.0712|4.0721|224.3243|1.4555|
> |**DSPO-Repaint-M**|0.1080|0.1868|**2.0663**|**4.0585**|223.3857|**1.4504**|
> ||||||||
> |**DPS**|0.0969|0.1757|1.9230|3.7675|193.4167|1.3564|
> |**Align-DPS-M**|0.0943|0.1734|1.9070|3.7287|189.6935|1.3428|
> |**IPO-DPS-M**|0.0937|**0.1732**|**1.9045**|3.7275|189.5445|**1.3412**|
> |**DSPO-DPS-M**|**0.0932**|0.1735|1.9062|**3.7269**|**189.2176**|1.3418|
>
> We will revise the manuscript to more explicitly state that this framework is a key contribution and will include these new experimental results to strengthen our claims.
>
> ___
> > **Response to Weakness 3:** Comparisons with state-of-the-art methods for the forecast or those methods that can be applied to this task.
>
> We thank the reviewer for this helpful comment. Our contribution is to utilize Preference Optimization to enhance Score-based DA rather than establishing a new forecast model to compare existing state-of-the-art methods.
>
> Consequently, our experimental design focuses on controlled ablation studies of score‑based DA methods to identify the effect of alignment.  In our experiments,  the frozen forecasting model (FengWu) serves two distinct roles: First, it generates the 48-hour background field that acts as the common input for all DA methods. Second, we use FengWu to assess forecast skill by running 48‑hour forecasts initialized from each analysis field.
>
> We have tried to use other forecast models, such as GraphCast or Pangu-Weather, but we meet practical constraints like the lack of access to them, at the required 1.4-degree resolution. Nevertheless, we firmly believe our present, focused comparison is enough to state our contributions. We will clarify this positioning more explicitly in the experimental setup section of the revised manuscript.
>
> ___
> > **Response to Weakness 4:** Explain the DA background in detail
>
> We thank the reviewer for this constructive suggestion. In the revised manuscript, we will expand the introduction to include a more detailed explanation of DA in the context of numerical weather prediction, and highlight that traditional approaches often require extensive trial-and-error tuning to obtain satisfactory results.
>
> Here is what we revised in Line 30:
> "**In atmospheric science, data assimilation (DA) provides accurate initial conditions that are essential for reliable weather forecasting. It is also used to generate high-fidelity reanalysis datasets, which serve as a fundamental resource for climate research.**
>
> ___
> Thank you for your helpful comments again. We hope this response can solve your concern.
> ___
> **Reference:**
>
> [1] https://doi.org/10.1029/2023MS004178
>
> [2] https://doi.org/10.1002/psp4.12588
>
> [3] https://doi.org/10.1016/j.jocs.2020.101171
>
> [4] https://doi.org/10.1016/j.geoen.2024.213275
>
> [5] https://arxiv.org/abs/2505.05452
>
> [6] https://arxiv.org/abs/2310.12036
>
> [7] https://arxiv.org/abs/2504.15176 ICLR2025

---

> ### Comment · Area_Chair_Foqp · 2025-08-06
>
> Please respond to the authors' rebuttal and indicate if you are happy to reconsider your score based on the rebuttal and the other reviewers' comments.

---

> ### Author Response · Authors · 2025-08-06
> **Looking forward to hearing from you**
>
> Dear Reviewer 46nx,
>
> Thank you for your thoughtful feedback again.
>
> To respond to your valuable comments, we have clarified how Align-DA's preference alignment paradigm is fundamentally different from sequential RL-DA approaches. We also elaborated on how this framework can be generalized beyond atmospheric science. Crucially, to provide strong empirical evidence for our claims, we performed new experiments. First, we conducted a preference optimization without physics-based rewards explicitly. The physical consistency improvement indicates the model is not simply overfitting on metrics, which addresses the "teaching to the test" concern directly. Second, to further validate that our alignment concept is a powerful and generalizable tool, we also demonstrated its success with other preference optimization methods, including IPO and DSPO.
>
> We have incorporated these new results and other clarifications into the manuscript and hope these additions fully address your concerns. We would be grateful for the chance to discuss this further with you.
>
> Best regards
>
> The authors

---

> ### Comment · Reviewer_46nx · 2025-08-08
>
> Thank you for your clarification and the additional experiments. After reading the reply, I now understand the significance of the contribution better.
>
> Ultimately, I consider there two ways to position this work: (1) A general framework to improve existing methods' performance at a specific task by incorporating new, domain-specific reward functions. (2) A new method for weather forecasting providing superior accuracy. At this point I believe the authors are more inclined toward (1); however, if this is the case, multiple small case studies demonstrating the effectiveness and generalizability of the proposed framework must be shown, where the atmospheric application studied in this paper can only be considered as one of them. If (2) is the case, comparison to existing weather forecasting approaches is necessary. In a similar vein, Reviewer eG1A also raised the concern of baseline comparison to traditional data assimilation.
>
> As critical items are still missing for either interpretation, I am refrained from raising my rating. Nonetheless, it appears to me that the proposed framework should be able to transfer to other domains, as the authors suggested, even though I have not seen any evidence. Plus, all other reviewers have given positive ratings. Therefore, although I am more toward keeping my borderline reject recommendation, I acknowledge the paper's potential value to the community and understand it could still be considered for acceptance with appropriate revisions.
>
> I apologize for the narrower time frame, but I am still glad to know the authors' thought about my concern.

---

> > ### Author Response · Authors · 2025-08-08
> >
> > Dear Reviewer 46nx,
> >
> > Thank you again for your thoughtful feedback. We are encouraged that our previous response helped you to "understand the significance of the contribution better." We fully understand your remaining concerns about the paper's positioning, and we appreciate this opportunity to provide a final clarification.
> >
> > You have viewed our work's positioning as a choice between two paths: (1) a general framework that requires multi-domain validation, or (2) a new weather forecasting method that requires comparison with SOTA forecasting models. However, we respectfully suggest that our work is to **introduce a novel and paradigm-shifting methodology to solve a critical, long-standing problem within a specific, high-impact scientific domain, that is data assimilation (DA) at the weather scale.**
> >
> > To address your concern, we would like to clarify that our paper is not just a "single simple case study". Atmospheric DA is a classic ill-posed problem, characterized by high dimensionality, sparse observations, and strong non-linear dynamics, which is a grand challenge. The quality of DA directly affects the downstream weather forecasts. Our work offers a novel solution to this specific bottleneck by replacing manual tuning with the preference alignment framework.
> >
> > We agree that the core mechanism of Align-DA has the potential to generalize to other domains. However, applying a new method to any complex scientific field requires deep domain expertise, significant engineering effort, and extensive data curation. We believe our paper will inspire people to perform preference alignment in their domain. Nevertheless, we have verified the generalization in three respects, including different guidance methods, any designed rewards (in response to Reviewer eG1A), and various optimization algorithms (in our first reply).
> >
> > Regarding the second position of our paper, as you suggested, we would like to emphasize again that **Align-DA is not an end-to-end forecasting model; it focuses on the pluggable data assimilation module.** Its relationship with models like Pangu-Weather or FengWu is **complementary**, not competitive. In our experimental design, a forecast model like FengWu serves two distinct roles: it is a fixed **background generator** and an **evaluation tool**.
> >
> > Our scientific question is: *Given the same fixed forecast model, can Align-DA produce a better initial state from sparse observations than other DA methods?* We acknowledge that the comparisons against traditional DA methods are valuable, and we have done them (in response to Reviewer eG1A). The results clearly show that Align-DA achieves superior forecast skill and physical consistency, powerfully demonstrating the advantages of our new paradigm over established methods.
> >
> >
> > In summary, we introduce preference alignment as a novel paradigm for DA. To validate this paradigm shift, we have:
> >
> > 1. **Addressed the "teaching to the test" concern** with a new experiment (aligning without physics rewards) to prove that our model learns the underlying physical reality, not just overfitting to metrics.
> >
> > 2. **Rigorously validated its components** through extensive ablation studies and comparisons with other alignment algorithms (IPO, DSPO).
> >
> > 3. **Extended it to more rewards** via a new experiment with an additional physical reward (hydrostatic balance).
> >
> > We are sincerely grateful for your rigor and insightful critiques. We hope this clarification successfully explains the scope of our work. We believe Align-DA holds significant potential and value for the AI for Science community.
> >
> > Thank you once again for your time and consideration.
> >
> > Sincerely,
> >
> > The Authors

---

> > > ### Comment · Reviewer_46nx · 2025-08-09
> > >
> > > While you attempt to differentiate your work from my concerns, I still do not see how you position your approach such that it offers a clear, substantial contribution. Simply answering the question ("Can Align-DA produce a better initial state from sparse observations than other DA methods?") is too narrow and does not address the broader impact. Moreover, based on the comparison with traditional DA methods in your response to Reviewer eG1A, I do not agree that Align-DA has demonstrated clear superiority. Therefore, I maintain my original recommendation, though I understand this may not affect the overall outcome given the other reviewers' feedback.

---

> > > > ### Author Response · Authors · 2025-08-09
> > > >
> > > > Dear Reviewer 46nx,
> > > >
> > > > Thank you for your further comments. We appreciate the chance to address your remaining concerns. We hope to provide more details from AI for Science to clarify our contributions.
> > > >
> > > > We would like to re-clarify our research scope. The main goal of our work—to produce a better initial state for weather forecast—is precisely the defining challenge of the entire field of data assimilation[1,2,3]. We would like to further clarify that improving this initial state is not an incremental step; it is a high-impact contribution that directly addresses a critical bottleneck in atmospheric science.
> > > >
> > > > In this research area, the assimilation accuracy is a preliminary pursuit. We may care more about the forecast skill and physical consistency. It is usually that the **previous studies focused on the assimilation accuracy only** (including the traditional 3D-Var). The underlying difficulties, such as expensive training, complex trade-offs, and time-consuming manual tuning, limit these methods from incorporating the forecast skill and physical consistency. Our framework introduces **a preference alignment** mechanism into **score-based DA** to tackle it. Our results and additional experiments strongly suggest the success of our framework. Meantime, our approach shows strong generalizability and we have validated it from three respects (different guidance methods, any designed rewards, and various optimization algorithms). This is our main contribution. Therefore,  we believe our work has a broad impact.
> > > >
> > > > To our knowledge, previous SDA papers have not been rigorously compared against traditional DA, like 3D-Var [4,5,6,7].  Nevertheless, we conduct such experiments because we aim to push Align-DA toward practical application. We would like to re-clarify the interpretation of these results. As we explained before (third paragraph), it is not a surprise that 3D-Var achieves a marginally better DA Accuracy (MSE), which is optimized specifically. However, this single metric does not capture the full performance. In atmospheric science, the ultimate measure of a DA system's quality is its impact on **the downstream forecast skill and the physical consistency** of the resulting analysis. This is precisely where our contribution lies. As our results clearly show, Align-DA demonstrates a substantial and significant improvement over 3D-Var in both Forecast Skill (MSE) and Physical Consistency (Geo-Score) (e.g., improving the Geo-Score from 88.34 to 94.89). For a large-scale atmospheric system, such gains are considered highly significant and directly translated to more reliable weather predictions.
> > > >
> > > > In summary, we sincerely hope we have better articulated that our work tackles a foundational problem with broad impact and that its advantages over traditional methods are clear.
> > > >
> > > > Thank you once again for your time.
> > > >
> > > > Sincerely,
> > > >
> > > > The Authors
> > > >
> > > > [1] Yi Xiao, et al. VAE-Var: Variational Autoencoder-Enhanced Variational Methods for Data Assimilation in Meteorology. ICLR2025
> > > >
> > > > [2] Kun Chen, et, al. FNP: Fourier Neural Processes for Arbitrary-Resolution Data Assimilation. NIPS2024
> > > >
> > > > [3] François Rozet et al.  Score-based Data Assimilation NIPS 2023
> > > >
> > > > [4] François Rozet et al.  Score-based Data Assimilation for a Two-Layer Quasi-Geostrophic Model.
> > > >
> > > > [5] Langwen Huang, et al. DiffDA ICML2024.
> > > >
> > > > [6] Peter Manshausen, et al. Generative Data Assimilation of Sparse Weather Station Observations at Kilometer Scales
> > > >
> > > > [7] Yongquan Qu, et, al. Deep Generative Data Assimilation in Multimodal Setting. CVPR2024.

---

### Author Response · Authors · 2025-08-07
**General Response**

---



**Dear Reviewers, PCs, ACs, and SACs,**

We sincerely thank you for your thorough, insightful, and constructive reviews.

___

We are greatly encouraged by the recognition of our contributions across multiple aspects:

* Reviewer **46nx** recognized that our framework is "well motivated" and that applying reinforcement learning to DA is "novel and original".
* Reviewer **oCk2** highlighted our work as the "first (to my knowledge) to adapt Direct Preference Optimization (DPO) to atmospheric data assimilation" and noted the "consistent improvements" shown in our experiments.
* Reviewer **eG1A** appreciated our "novel application effort" of multi-score DPO on climate science data and the paper's "well-written and structured".
* Reviewer **r9ir** valued our "thorough" exploration of multiple objectives and the "detailed derivations of the mathematical formulation".
* Reviewer **ywbS** found the core idea of applying DPO with multiple rewards "novel" and acknowledged the "consistent performance gains" and "technical rigor" of our work.

___

In response to your valuable feedback, we have undertaken the following **clarifications and improvements**:

* **More DA backgrounds:** Following the suggestions from Reviewers **46nx** and **oCk2**, we have incorporated a more detailed explanation of DA in our revised version.

* **Core Contribution & Novelty:** To clarify the novelty further as suggested by Reviewers **46nx** and **eG1A**, we have refined the introduction to more clearly distinguish **Align-DA** from previous sequential RL-based DA efforts. We also emphasize that our core contribution is a **paradigm shift**, not a new model structure.

* **Empirical Validation with New Experiments:** To address key questions from all reviewers, we conducted several new experiments:
  *   **Addressing "Teaching to the Test" (Reviewer 46nx):** We ran a new experiment where the model was aligned *without* explicit physics-based rewards. The model still demonstrated significant improvements in physical consistency, proving it learns a fundamentally realistic system state rather than merely overfitting to the reward metrics.
  *   **Comparison with Traditional Methods (Reviewer eG1A):** As requested, we performed a new comparison against a traditional **3D-Var** baseline. The results show that Align-DA achieves better forecast skill and physical consistency, demonstrating its advantages over established methods.
  *   **Expanding Physics Constraints (Reviewer eG1A):** To showcase the framework's flexibility, we incorporated an additional physics constraint (hydrostatic balance) and confirmed that it leads to further performance gains.
  *   **Varying Observation Densities (Reviewers r9ir, ywbS):** We re-emphasized the extensive ablation studies with observation densities from 1% to 10%. The results consistently show that our alignment method outperforms baselines across all densities.
  *   **Statistical Significance (Reviewer ywbS):** We performed statistical significance testing, confirming that the performance gains from Align-DA are substantial and not due to random chance.

* **Improved Clarity, Positioning, and Formatting:**
  *   As suggested by Reviewers **oCk2** and **eG1A**, we have clarified the role of the FengWu forecast model as a frozen component for generating backgrounds and providing a reward signal, a setup that highlights the unique learning capability of Align-DA.
  *   In response to Reviewers **r9ir** and **ywbS**, we have added a detailed description of the computational costs, demonstrating the efficiency of our alignment process.
  *   We have carefully addressed all formatting, citation, and clarity suggestions from Reviewer **ywbS** to improve the paper's readability.

We hope that these comprehensive revisions and new experiments may address the points raised during the review process.

___

Finally, we would like to re-emphasize the key contributions of our work:

*   Align-DA, a novel and scalable preference alignment framework that replaces the empirical tuning DA.
*   A flexible method for incorporating multiple, complex rewards—including analysis accuracy, downstream forecast skill, and physical consistency.
*   Extensive new empirical evidence demonstrating that Align-DA is robust, statistically significant, and provides benefits over both non-aligned and traditional baselines.

___

With two days remaining in the discussion stage, we welcome any further feedback and are delighted to continue the discussion.


Best regards,

The Authors

---

### Public Comment · ~Jing-An_Sun1 · 2025-11-07
**Reply to Official comment by PCs**

Dear Program Chairs,

Thank you very much for sharing the additional review from Reviewer 918U with us. We sincerely appreciate your commitment to transparency and fairness throughout the review process.

We would also like to extend our sincere thanks to Reviewer 918U for their time and thoughtful comments. We are encouraged that they found our work to be novel, sound, and of interest to the community.

Although the formal discussion period has concluded, we would like to briefly address the valuable points raised. Much of this was discussed with other reviewers, and we have already incorporated relevant details into our camera-ready paper.

- **On Computational Cost and Baselines:** We now provide a detailed computational cost breakdown, confirming the **DPO alignment stage is highly efficient**. Our baselines (Repaint and DPS) were chosen as key score-based DA methods in our paper.
- **On Why Physics-Only Alignment Can Slightly Worsen Performance:** This highlights a core finding: the initial single-level physical reward (geostrophic balance) is an idealized constraint that can conflict with other objectives, underscoring that **multi-reward alignment is crucial** for robust gains. Our follow-up experiments with a multi-level physical constraint (hydrostatic balance) did yield overall improvement even with single-reward optimization.
- **On Suggestions for Clarity:** We have addressed similar feedback from other reviewers by revising figure captions and clarifying definitions of terms like ERA5 and ObsFree in the camera-ready version.

Once again, thank you for your hard work in managing this conference. Please extend our gratitude to Reviewer 918U as well.

Sincerely,

The Authors of Submission 2512

---

### Note · Authors · 2025-08-14

We sincerely thank all reviewers, ACs, and SACs for their insightful feedback and constructive suggestions. We are encouraged that reviewers widely recognize the core novelty and technical rigor of our work.

During the discussion phase, we addressed the raised concerns with concrete clarifications and updates, including:

- **Core Contribution & Novelty:** We clarified that Align-DA introduces a new **paradigm** for data assimilation (DA), where the flexible preference alignment framework replaces empirical manual tuning.
- **Empirical** **Validation** **& Baselines:** In response to reviewer feedback, we conducted several new experiments. This included: (1) A direct comparison against traditional **3D-Var**, which demonstrated Align-DA's superior performance in forecast skill and physical consistency. (2) An experiment aligning the model **without explicit physics-based rewards** to address the "teaching to the test" concern. (3) The addition of a **new physics constraint** (hydrostatic balance) to showcase the framework's **flexibility**.
- **Generality & Robustness:** We demonstrated the robustness of Align-DA through extensive ablation studies across various **observation densities** (from 1% to 10%). We also validated that our alignment concept is generalizable by successfully implementing it with other preference optimization algorithms, including **IPO** **and DSPO**.
- **Clarity & Discussion:** We have revised the manuscript to incorporate a more detailed background on DA, an analysis of computational overhead, and **statistical significance testing** for our results, significantly improving the paper's clarity and completeness as suggested.

These clarifications and new experiments emphasize the novelty, robustness, and generalizability of Align-DA. By introducing a novel, data-driven preference alignment paradigm, our work offers a promising new direction for overcoming the long-standing bottleneck of manual tuning in atmospheric DA.

We believe Align-DA addresses a foundational problem in atmospheric science, offering a pluggable alignment module that bridges modern AI techniques with complex scientific challenges and constructs a solid foundation for future research in the AI for Science domain.

Once again, we sincerely thank you for your time and effort during this review.

Yours sincerely,

The Authors of Submission 2512

---

### Decision · Program_Chairs · 2025-09-17

**Decision:**

Accept (poster)

**Comment:**

Paper summary:

This paper presents a reinforcement learning-based framework for data assimilation (DA), named Align-DA, to estimate the system states of atmospheric applications given their prior states and observations. In this framework, diffusion models are used for DA, and goals of assimilation and domain-specific knowledge can be formulated as reward functions that guide the learning process of the models. As an example, the authors explored regularizing the optimization with three metrics, assimilation accuracy, forecast skill, and physical adherence. Experimental results demonstrate that models trained using the proposed framework with multiple rewards outperform those trained without or with a single reward.

Summary of the discussion:

4 reviewers for for acceptance. 2 with confidence and 2 only weakly. 1 reviwer votes weakly for rejection.

The only reviewer voting for rejection said they cannot see that the authors are positioning the paper such that it offers a clear, substantial contribution. The clarified core research problem appears to be narrower than necessary given the other materials presented. Comparing to an existing DA method, the proposed Align-DA has not demonstrated clear superiority.

None of the other 4 reviewers shared this concern, which reduces the weight of the latter comment.

Recommendation:

This is a borderline paper with 4 reviewers voting for acceptance (two weakly) and another reviwer voting weakly for rejection. Based on the above, I recommend accepting the paper and encourage the authors to use the feedback provided to improve the paper for its camera ready version.